# Go With the Flow: Fast Diffusion for Gaussian Mixture Models

**George Rapakoulias** [1] [*]    **Ali Reza Pedram** [1,2]    **Fengjiao Liu** [3]    **Lingjiong Zhu** [4]

**Panagiotis Tsiotras** [1]

[1] Department of Aerospace Engineering, Georgia Institute of Technology, Atlanta, GA
[2] School of Computer Science, University of Oklahoma, Norman, OK
[3] Department of ECE, FAMU-FSU College of Engineering, Tallahassee, FL
[4] Department of Mathematics, Florida State University, Tallahassee, FL

## Abstract

Schrödinger Bridges (SBs) are diffusion processes that steer, in finite time, a given initial distribution to another final one while minimizing a suitable cost functional. Although various methods for computing SBs have recently been proposed in the literature, most of these approaches require computationally expensive training schemes, even for solving low-dimensional problems. In this work, we propose an analytic parametrization of a set of feasible policies for steering the distribution of a dynamical system from one Gaussian Mixture Model (GMM) to another. Instead of relying on standard non-convex optimization techniques, the optimal policy within the set can be approximated as the solution of a low-dimensional linear program whose dimension scales linearly with the number of components in each mixture. The proposed method generalizes naturally to more general classes of dynamical systems, such as controllable linear time-varying systems, enabling efficient solutions to multi-marginal momentum SBs between GMMs, a challenging distribution interpolation problem. We showcase the potential of this approach in low-to-moderate dimensional problems such as image-to-image translation in the latent space of an autoencoder, learning of cellular dynamics using multi-marginal momentum SBs, and various other examples. The implementation is publicly available at `https://github.com/georgeRapa/GMMflow`.

## 1 Introduction and Background

The problem of finding mappings between distributions of data, originally known as the *Optimal Transport* (OT) problem in mathematics, has received significant attention in recent years in multiple research fields, due to its application in problems such as generative AI (Ruthotto & Haber, 2021; Arjovsky et al., 2017), biology (Bunne et al., 2023b; Bunne & Rätsch, 2023; Tong et al., 2020), mean field problems (Liu et al., 2022) and control theory (Chen et al., 2015a,b; Rapakoulias & Tsiotras, 2024) among many others. Despite appearing static in nature, reformulating OT in the context of dynamical systems imbues it with further structure and unlocks tools from the literature on dynamical systems that can be employed for its efficient solution (Benamou & Brenier, 2000).

To set the stage, consider two distributions $\rho_0, \rho_1$, supported on the $d$-dimensional Euclidean space, denoted by $\mathbb{R}^d$, and consider the regularized version of the static OT optimization problem, known as the *Entropic Optimal Transport* (EOT) problem (Peyré & Cuturi, 2019):

$$\min_{\pi \in \Pi(\rho_0, \rho_1)} \int_{\mathbb{R}^d \times \mathbb{R}^d} \frac{1}{2} \|x_0 - x_1\|^2 \mathrm{d}\pi(x_0, x_1) - \epsilon H(\pi), \tag{1}$$

---

[*]Corresponding author: `grap@gatech.edu`

where $\pi(x_0, x_1)$ is the transport plan (also referred to as coupling) between $\rho_0, \rho_1$, $\Pi(\rho_0, \rho_1)$ is the set of all joint distributions with marginals $\rho_0, \rho_1$, and $H$ is the differential entropy, defined by $H(\rho) \triangleq -\int \rho(x) \log \rho(x) \, dx$. The corresponding dynamic formulation of the EOT problem is known as the *Schrödinger Bridge Problem* (SBP) (Léonard, 2014; Chen et al., 2021). When formulated as a stochastic optimal control problem, the SBP is given by

$$\min_{u \in \mathcal{U}} \ J_{\text{SB}} \triangleq \mathbb{E}_{x_t \sim \rho_t} \left[ \int_0^1 \frac{1}{2\epsilon} \|u_t(x_t)\|^2 dt \right], \quad \text{s.t.} \begin{cases} dx_t = u_t(x_t) \, dt + \sqrt{\epsilon} \, dw, \\ x_0 \sim \rho_0, \quad x_1 \sim \rho_1, \end{cases} \tag{2}$$

where the objective is to find an optimal drift function $u_t(x_t)$, also referred to as the control policy in the context of control applications, belonging to a set of adapted finite-energy policies $\mathcal{U}$, such that, when applied to the stochastic dynamical system defined by the first constraint in (2), the marginal distribution specified in the second constraint is guaranteed, i.e., for initial conditions sampled at time $t = 0$ from $\rho_0$, the state at time $t = 1$ will be distributed according to $\rho_1$, and the cost $J_{\text{SB}}$ in (2) will be minimized.

The increased practical applications of EOT and SBs in multiple machine learning problems, especially in high-dimensional generative applications where the boundary distributions $\rho_0, \rho_1$ are only available through a finite number of samples, have led to the development of a multitude of algorithms over recent years. The state-of-the-art methods for solving SBs leverage the properties of problem (2), such as the decomposition of the optimal probability flow into conditional problems that are easier to solve, sometimes even analytically (Chen et al., 2016; Lipman et al., 2023; Liu et al., 2023). In this category of methods, a recent technique known as *Diffusion Schrödinger Bridge Matching* (DSBM) (Shi et al., 2023; Peluchetti, 2023), or its deterministic counterpart, known as *Flow Matching* (FM) (Lipman et al., 2023) or *Rectified Flow* (RF) (Liu et al., 2023), leverages the decomposition of the optimal probability flow to a mixture of flows conditioned on their respective endpoints and retrieves an approximation of the optimal solution to (2) as a mixture of conditional policies that are easy to calculate. Theoretically, one needs to combine an infinite number of conditional flows to retrieve the true flow, due to the continuous support of the boundary distributions. To overcome this issue, a neural network is usually trained to approximate this infinite mixture.

While the DSBM and the various Flow Matching algorithms have proven very effective in high-dimensional problems, the efficient solution of SBs in simpler problems is hindered by the lack of closed-form expressions in all but very few special cases with Gaussian marginal distributions (Bunne et al., 2023a). To tackle this problem and avoid costly neural network training in smaller problems, recent methods such as Light-SB (LSB) (Korotin et al., 2024) and Light-SB Matching (LSBM) (Gushchin et al., 2024) have been proposed to obtain quick and efficient solutions to SBs within seconds, for problems with low-complexity boundary distributions, such as mixture models. These methods work by an efficient parametrization of the Schrödinger potentials, a key component of the SB. Because this parametrization does not lead to closed-form expressions for the boundary distributions of the SB, the calculation of its parameters is carried out through optimization.

Inspired by the flow decomposition idea behind DSBM and FM methods, and motivated by the need to obtain light-weight and fast SB solvers for a wide class of SB problems, in this paper, we solve the problem of finding a policy that can efficiently steer the distribution of a dynamical system from a *Gaussian Mixture Model* (GMM) to another one, using a mixture of conditional policies that can each steer the individual components of the initial mixture to the components of the terminal mixture. This approach, which is tailored to GMMs, separates the problem of fitting the boundary distributions to the data and solving the SB, resulting in improved accuracy with regard to the marginal distribution fitting. More specifically, we claim the following **main contributions**:

1. We present a computationally efficient, training-free method to solve the Schrödinger Bridge and the multi-marginal Momentum Schrödinger Bridge problems in the case where the boundary distributions are Gaussian Mixture Models.

2. In contrast to existing approaches, our method can handle both stochastic and deterministic versions of the problem (2). Based on a control-theoretic formulation, our approach also naturally generalizes to dynamical systems with a general Linear Time-Varying (LTV) structure, with the control input and stochastic component having different dimensions than the state, which could be of interest in Mean Field Games (MFG), multi-agent control applications (Ruthotto et al., 2020; Liu et al., 2022; Chen, 2024), and higher order distribution interpolation such as Wasserstein splines (Chen et al., 2018).

3. We demonstrate the substantial potential of our algorithm in low-dimensional problems, moderate-dimensional image-to-image translation tasks, and multi-marginal diffusion learning problems. Specifically, we show that our approach outperforms state-of-the-art lightweight methods for solving the SB problem both in terms of training speed and accuracy of the learned boundary distributions, when these are available through samples (40% better FID scores in the image translation task and one order of magnitude better MMD scores in the multi-marginal diffusion-learning problems).

4. Finally, we extend our method to problems with continuous GMM marginal distributions, a wide class of distributions that can capture multiple useful distributions with heavy tails, and we use our approach to construct upper bounds on the 2-Wasserstein distance and approximate the displacement interpolation between Student-t distributions.

## 2 Preliminaries

### 2.1 Diffusion Schrödinger Bridge Matching and Flow Matching

The composition of diffusion processes as mixtures of processes conditioned on their endpoints was originally proposed by Peluchetti (2021) as a simulation-free algorithm for generative modeling applications. The concept was later tailored to solve the SBP in the DSBM algorithm (Shi et al., 2023), proposed concurrently by Peluchetti (2023). Similar simulation-free methods have also been proposed to solve variants of the same problem in Albergo & Vanden-Eijnden (2023) and in Liu et al. (2024); Theodoropoulos et al. (2025) for the stochastic bridge setting, as well as in Liu et al. (2023) and Lipman et al. (2023) for the deterministic setting. For a more comprehensive overview along with comparisons with other available methods, we refer the reader to (Shi et al., 2023, Section 5) and (Peluchetti, 2023, Section 5).

Given problem (2), the main idea is to decompose the problem into a sequence of elementary conditional subproblems that are easier to solve, and then express the solution as a mixture of the solutions of the conditional subproblems. This idea has an intuitive motivation: Informally, finding a policy that transports the state distribution from an initial density to a target density can be separated into two problems. First, one needs to figure out a transport plan solving the "who goes where" problem and then one needs to compute a point-to-point optimal policy, that solves the "how to get there" problem (Terpin et al., 2024a). In many cases, the two subproblems are decoupled (Chen et al., 2021, 2016; Terpin et al., 2024a); most importantly, however, computing the point-to-point optimal policy can be solved analytically for simple dynamical systems, such as the one in (2).

More precisely, the optimal probability flow $\rho_t^*$ of Problem (2) is known (Föllmer, 1988; Chen et al., 2021) to admit the decomposition

$$\rho_t^*(x) = \int_{\mathbb{R}^d \times \mathbb{R}^d} W_{t|x_0,x_1}(x) \, \mathrm{d}\pi_\epsilon^*(x_0, x_1), \tag{3}$$

where $W_{t|x_0,x_1}(x)$ is the probability density of the unforced dynamics $\mathrm{d}x_t = \sqrt{\epsilon}\,\mathrm{d}w$, namely the Brownian motion kernel, pinned at $x_0$ for $t = 0$ and at $x_1$ for $t = 1$, and $\pi_\epsilon^*(x_0, x_1)$ is the entropic optimal transport plan between $\rho_0, \rho_1$ solving (1). Dai Pra (1991) showed that $W_{t|x_0,x_1}(x)$ solves the following optimal control problem

$$\min_{u_{t|0,1} \in \mathcal{U}} \quad J_{0,1}(x_0, x_1) \triangleq \mathbb{E}\left[\int_0^1 \|u_{t|0,1}(x_t)\|^2 \mathrm{d}t\right], \quad \text{s.t.} \begin{cases} \mathrm{d}x_t = u_{t|0,1}(x_t)\,\mathrm{d}t + \sqrt{\epsilon}\,\mathrm{d}w, \\ x_0 \sim \delta_{x_0}, \quad x_1 \sim \delta_{x_1}, \end{cases} \tag{4}$$

where $\delta_{x_0}, \delta_{x_1}$ are Dirac delta functions centered on $x_0$ and $x_1$, respectively. Assuming $\rho_{t|0,1}(x)$ and $u_{t|0,1}(x)$ solve (4), one can construct a feasible solution for the original problem (2) using any transport plan $q(x_0, x_1) \in \Pi(\rho_0, \rho_1)$, i.e., any joint distribution between the desired boundaries $\rho_0, \rho_1$, using the mixtures

$$\rho_t(x) = \int_{\mathbb{R}^d \times \mathbb{R}^d} \rho_{t|0,1}(x) q(x_0, x_1) \, \mathrm{d}x_0 \, \mathrm{d}x_1, \tag{5a}$$

$$u_t(x) = \int_{\mathbb{R}^d \times \mathbb{R}^d} u_{t|0,1}(x) \frac{\rho_{t|0,1}(x) q(x_0, x_1)}{\rho_t(x)} \, \mathrm{d}x_0 \, \mathrm{d}x_1. \tag{5b}$$

Showing that the flow (5a) is a feasible solution to (2) for any valid coupling $q(x_0, x_1)$ amounts to verifying that the flow $\rho_t(x)$ satisfies the boundary distributions $\rho_0, \rho_1$ at times $t = 0$ and $t = 1$,

respectively. To prove that the policy (5b) produces (5a), it suffices to show that the pair (5a), (5b) satisfies the FPK PDE (Lipman et al., 2023; Liu et al., 2024). When $q(x_0, x_1) = \pi^*_\epsilon(x_0, x_1)$, (5a) reduces to (3), and (5b) recovers the optimal solution to (2) (Shi et al., 2023; Peluchetti, 2023).

## 2.2 Schrödinger Bridges with Gaussian Marginals

The SBP with Gaussian Marginals, henceforth referred to as the Gaussian SB (GSB), has been extensively studied in the literature and can be solved either analytically for simple choices of prior dynamics (Bunne et al., 2023a) or as a convex semidefinite optimization problem for general linear dynamical systems both for continuous and discrete time cases (Chen et al., 2015b; Liu et al., 2025). Because we use the GSB as a building block to construct a policy that works with general GMM boundary distributions, we briefly review the available methods for its solution here. To this end, consider the optimization problem with Gaussian marginals

$$\min_{u \in \mathcal{U}} \ J_{\text{GSB}} \triangleq \mathbb{E}\left[\int_0^1 \|u_t(x_t)\|^2 \, dt\right], \quad \text{s.t.} \begin{cases} dx_t = u_t(x_t) \, dt + \sqrt{\epsilon} \, dw, \\ x_0 \sim \mathcal{N}(\mu_0, \Sigma_0), \quad x_1 \sim \mathcal{N}(\mu_1, \Sigma_1), \end{cases} \quad (6)$$

where $\mu_0, \Sigma_0, \mu_1, \Sigma_1$ are the means and covariances of the initial and final Gaussian boundary distributions, respectively.

**Proposition 1.** *(Bunne et al., 2023a, Theorem 3) The optimal solution to Problem (6) is given by* $u_t(x) = K_t(x - \mu_t) + v_t$ *with* $\mu_t = (1-t)\mu_0 + t\mu_1$, $v_t = \mu_1 - \mu_0$, *and* $K_t = S_t^\mathsf{T}\Sigma_t^{-1}$, *where*

$$\Sigma_t = (1-t)^2\Sigma_0 + t^2\Sigma_1 + (1-t)t(C_\epsilon + C_\epsilon^\mathsf{T} + \epsilon I),$$
$$S_t = t(\Sigma_1 - C_\epsilon^\mathsf{T}) - (1-t)(\Sigma_0 - C_\epsilon) - \epsilon t I,$$

*with* $C_\epsilon = \frac{1}{2}(\Sigma_0^{\frac{1}{2}} D_\epsilon \Sigma_0^{-\frac{1}{2}} - \epsilon I)$ *and* $D_\epsilon = (4\Sigma_0^{\frac{1}{2}}\Sigma_1\Sigma_0^{\frac{1}{2}} + \epsilon^2 I)^{\frac{1}{2}}$.

Furthermore, the optimal value of the cost $J_{\text{GSB}}$ in (6) is given by the following proposition.

**Proposition 2.** *Consider Problem (6) with* $\epsilon > 0$. *Then, the optimal value for the cost* $J_{\text{GSB}}$ *is*

$$J_{\text{GSB}} = \|\mu_1 - \mu_0\|^2 + \text{tr}(\Sigma_0) + \text{tr}(\Sigma_1) - \epsilon\left(\text{tr}M_{2\epsilon} - \log\det M_{2\epsilon} + \log\det\Sigma_1\right) + c, \quad (8)$$

*where* $M_\epsilon = I + (I + \frac{16}{\epsilon^2}\Sigma_0\Sigma_1)^{\frac{1}{2}}$, *and* $c$ *is a constant independent of the boundary distributions.*

In the limit of $\epsilon \to 0$, (8) reduces to the well known Bures-Wasserstein distance (Bhatia et al., 2019), defined by

$$\text{BW}(\mathcal{N}(\mu_0, \Sigma_0)\|\mathcal{N}(\mu_1, \Sigma_1)) \triangleq \|\mu_1 - \mu_0\|^2 + \text{tr}(\Sigma_0) + \text{tr}(\Sigma_1) - 2\text{tr}\left(\Sigma_1^{\frac{1}{2}}\Sigma_0\Sigma_1^{\frac{1}{2}}\right)^{\frac{1}{2}}. \quad (9)$$

## 2.3 Momentum Schrödinger Bridges with multiple Gaussian marginals

Other than Problem (6), we will also make use of the solution to the corresponding Gaussian multi-marginal Momentum SB (GMSB) (Chen et al., 2019), which is a variation of Problem (6) where the dynamics include a momentum term, while the goal is to match multiple marginal distributions at regular time intervals. More specifically, the GMSB problem reads

$$\min_{u \in \mathcal{U}} \quad \mathbb{E}\left[\int_0^1 \|u_t(x_t, v_t)\|^2 \, dt\right], \quad (10a)$$
$$\text{s.t.} \quad dx_t = v_t \, dt, \quad dv_t = u_t(x_t, v_t) \, dt + \sqrt{\epsilon} \, dw, \quad (10b)$$
$$x_{t_i} \sim \mathcal{N}(\mu_{t_i}, \Sigma_{t_i}), \quad i = 1, \ldots, N, \quad (10c)$$

where $N$ is the number of marginal constraints and the joint space of position and velocity, namely $x_t, v_t \in \mathbb{R}^d$, is referred to as the phase space. Compared to the standard SB problem, and apart from having multiple marginal distributions, the GMSB problem only constrains the position component of the phase space, namely $x_t$. Even when the marginals are Gaussian, a closed-form solution to (10) is unknown; however, the problem can be solved efficiently using semidefinite programming. In the special case where the noise parameter $\epsilon$ is zero, the problem is known as the Gaussian Wasserstein spline Problem (Chen et al., 2018), and an efficient semidefinite formulation for solving it is given in Chen et al. (2018). Since the semidefinite formulation for solving (10) is a well-studied problem, due to space considerations, we defer it to Appendix D. Finally, we note that a generalization of the

Problems (6) and (10) is achieved by a Linear Time Varying (LTV) structure in the prior dynamics of the bridge, i.e., replacing the first constraint in (6) with

$$\mathrm{d}x_t = A_t x_t \, \mathrm{d}t + B_t u_t \, \mathrm{d}t + D_t \, \mathrm{d}w, \tag{11}$$

where $x_t \in \mathbb{R}^d$, $A_t \in \mathbb{R}^{d \times d}$, $u_t \in \mathbb{R}^m$, $B_t \in \mathbb{R}^{d \times m}$, $D_t \in \mathbb{R}^{d \times q}$, $w_t \in \mathbb{R}^q$. Bridges with prior dynamics of the form (11) have been extensively studied in the context of control theory, with the corresponding literature known as Covariance Steering (CS) (Chen et al., 2015a,b; Bakolas, 2018; Liu et al., 2025). CS problems can be formulated as convex programs for both continuous and discrete-time cases (Chen et al., 2015b; Liu et al., 2025), and therefore attain an efficient and exact calculation, which we will exploit in the sequel.

## 3 Fast Diffusion for Mixture Models

### 3.1 Gaussian Mixture Schrödinger Bridge

Equation (5b) expresses the policy of Problem (2) as an infinite mixture of conditional, point-to-point policies. In this section, we extend this idea to construct a mixture policy, consisting of conditional policies each solving a Gaussian bridge sub-problem of the form (6). To this end, consider the problem

$$\min_{u \in \mathcal{U}} \quad J_{\mathrm{GMM}} \triangleq \mathbb{E}\left[ \int_0^1 \|u_t(x_t)\|^2 \, \mathrm{d}t \right], \tag{12a}$$

$$\text{s.t.} \quad \mathrm{d}x_t = u_t(x_t) \, \mathrm{d}t + \sqrt{\epsilon} \, \mathrm{d}w, \tag{12b}$$

$$x_0 \sim \sum_{i=1}^{N_0} \alpha_0^i \mathcal{N}\left(\mu_0^i, \Sigma_0^i\right), \quad x_1 \sim \sum_{j=1}^{N_1} \alpha_1^j \mathcal{N}\left(\mu_1^j, \Sigma_1^j\right). \tag{12c}$$

The main result is summarized in the following theorem.

**Theorem 1.** *Consider problem (12), with $N_0$ components in the initial mixture and $N_1$ components in the terminal mixture. Assume that $u_{t|ij}$ is the conditional policy that solves the $(i,j)$-GSB problem, that is, the bridge from the $i$-th component of the initial mixture, to the $j$-th component of the terminal mixture and let the resulting probability flow be $\rho_{t|ij}$. Furthermore, let $\lambda_{ij} \geq 0$ such that, for all $j \in \{1, 2, \ldots, N_1\}$, $\sum_i \lambda_{ij} = \alpha_1^j$ and such that, for all $i \in \{1, 2, \ldots, N_0\}$, $\sum_j \lambda_{ij} = \alpha_0^i$. Then, the policy*

$$u_t(x) = \sum_{i,j} u_{t|ij}(x) \frac{\rho_{t|ij}(x)\lambda_{ij}}{\sum_{i,j} \rho_{t|ij}(x)\lambda_{ij}}, \tag{13}$$

*is a feasible policy for Problem (12), and the corresponding probability flow is*

$$\rho_t(x) = \sum_{i,j} \rho_{t|ij}(x)\lambda_{ij}. \tag{14}$$

The mixture policy (13) is a weighted average of conditional policies, weighted according to $\lambda_{ij}\rho_{t|ij}(x)$, while the denominator $\sum_{i,j} \rho_{t|ij}(x)\lambda_{ij}$ is just a normalizing constant. Since $\rho_{t|ij}(x)$ is a Gaussian distribution centered at the mean of the $(i,j)$-Gaussian bridge at time $t$, this weighting scheme prioritizes the conditional policies whose mean is closer to the value of $x$ at the time $t$.

Equation (13) provides a set feasible of solutions to Problem (12), for all values of $\lambda_{ij}$ satisfying the conditions of Theorem 1. Obtaining the optimal policy within this set is challenging, in general. Alternatively, we can formulate a tractable problem by minimizing an upper bound to the original minimum effort cost function (12a), which is linear with respect to the transport plan $\lambda_{ij}$. More formally, we have the following theorem.

**Theorem 2.** *Let $J_{ij}$ be the optimal cost of solving the $(i,j)$-Gaussian bridge subproblem of the form (6) with marginal distributions the $i$-th component of the initial and the $j$-th component of the terminal mixture. Then, the cost function of the linear optimization problem*

$$\min_{\lambda_{ij} \geq 0} \quad J_{\mathrm{OT}} \triangleq \sum_{i,j} \lambda_{ij} J_{ij} \tag{15a}$$

$$\text{s.t.} \quad \sum_j \lambda_{ij} = \alpha_0^i \ \ \forall i \in \{1, 2, \ldots, N_0\}, \text{ and } \sum_i \lambda_{ij} = \alpha_1^j, \ \ \forall j \in \{1, 2, \ldots, N_1\}, \tag{15b}$$

*provides an upper bound for (12a), i.e., $J_{\mathrm{GMM}} \leq J_{\mathrm{OT}}$, for all positive values of $\lambda_{ij}$ satisfying (15b).*

For clarity of exposition, we defer the proofs of Theorems 1, 2 to Appendix A, along with an optimality analysis of the upper bound of Theorem 2.

In practice, to use policy (13) in problems where the boundary distributions are available only through samples, GMMs are first fitted in the samples of $\rho_0, \rho_1$ using the Expectation Maximization (EM) algorithm (Bishop & Nasrabadi, 2006), and then (13) is calculated using Theorems 1, 2. For clarity of exposition, we present an overview of this approach in Algorithms 1, 2 for both training and inference, accompanied by the corresponding theoretical complexity analysis in Appendix A.6.

---

**Algorithm 1** GMMflow training

---

    **Input:** Samples from boundary distributions $\rho_0, \rho_1$; number of GMM components $N_0$ and $N_1$, noise level $\epsilon \geq 0$.

    $\{\alpha_0^i, \mu_0^i, \Sigma_0^i\}_{i=1}^{N_0} \leftarrow \text{EM}(\rho_0, N_0)$   *// fits a GMM to initial dataset*
    $\{\alpha_1^j, \mu_1^j, \Sigma_1^j\}_{j=1}^{N_1} \leftarrow \text{EM}(\rho_1, N_1)$   *// fits a GMM to final dataset*
    **for** $(i, j) \in \{1, \ldots, N_0\} \times \{1, \ldots, N_1\}$ **do**   *//compute in parallel*
        $\{u_{t|ij}, \rho_{t|ij}, J_{ij}\} \leftarrow \text{CS}(\mu_0^i, \Sigma_0^i, \mu_1^j, \Sigma_1^j)$   *//solves the $(i, j)$-th conditional GSB*
    **end for**
    $\lambda_{ij} \leftarrow \text{SOLVE (15) USING } \{J_{ij}, \alpha_0^i, \alpha_1^j\}$
    **return** $u_{t|ij}, \rho_{t|ij}, \lambda_{ij}$

---

**Algorithm 2** GMMflow inference

---

    **Input:** Component-level solutions $u_{t|ij}, \rho_{t|ij}$, transport plan $\lambda_{ij}$, Initial condition $x_0 \sim \rho_0$, SDE integrator $\texttt{sde\_int}()$.

    $u_t(x) \leftarrow$ Compute (13)
    $x_t \leftarrow \texttt{sde\_int}((12b), x_0, t \in [0, 1])$
    **return** $x_1$

---

### 3.2 Multi-Marginal Problems

In this section, we generalize the results of Section 3.1 to solve the multi-marginal momentum SBs problem (Chen et al., 2023, 2019) with GMM marginal distributions. That is, we solve

$$\min_{u \in \mathcal{U}} \quad J_{\text{GMM}} \triangleq \mathbb{E}\left[\int_0^1 \|u_t(x_t, v_t)\|^2 \, dt\right] \tag{16a}$$

$$\text{s.t.} \quad dx_t = v_t \, dt, \quad dv_t = u_t(x_t, v_t) \, dt + \sqrt{\epsilon} \, dw_t, \tag{16b}$$

$$x_{t_i} \sim \sum_{k=1}^{N_i} \alpha_i^k \mathcal{N}(\mu_i^k, \Sigma_i^k), \quad i = 1, \ldots M, \tag{16c}$$

where the $i$-th marginal mixture is assumed to have $N_i$ components. Similarly to Theorem 1, we will combine conditional GMSBs of the form (10) to build a feasible set of policies solving (16). To facilitate notation, we denote by $\mathbf{i} = (i_1, \ldots, i_M)$ the index of the conditional multi-marginal GMSB and use the notation $\{\mathbf{i}|i_j = k\}$ to denote the set of all values of $\mathbf{i}$ such that $i_j = k$. With this notation in mind, we provide the following generalizations of Theorems 1 and 2:

**Theorem 3.** *Consider problem (16), with $M$ marginal mixture distributions, each having $N_i$ Gaussian components, where $i = 1, \ldots M$. Let $\mathbf{i} = (i_1 \ldots i_M)$ be an $M$-dimensional index, $u_{t|\mathbf{i}}$ be the conditional policy that solves the $\mathbf{i}$-GMSB problem, that is, the Gaussian multi-marginal momentum Schrödinger Bridge going through the $(i_1, \ldots, i_M)$ components of the marginal mixture models, and let the resulting probability flow be $\rho_{t|\mathbf{i}}$. Furthermore, let $\lambda_{\mathbf{i}} \geq 0$ be such that, for all $j = 1, \ldots, M$ and for all $k = 1, \ldots, N_j$,*

$$\sum_{\{\mathbf{i}|i_j=k\}} \lambda_{\mathbf{i}} = \alpha_j^k. \tag{17}$$

*Then, the policy*

$$u_t(x, v) = \sum_{\mathbf{i}} u_{t|\mathbf{i}}(x, v) \frac{\rho_{t|\mathbf{i}}(x,v)\lambda_{\mathbf{i}}}{\sum_{\mathbf{i}} \rho_{t|\mathbf{i}}(x,v)\lambda_{\mathbf{i}}}, \tag{18}$$

*is a feasible policy for Problem (12), and the corresponding probability flow is*

$$\rho_t(x, v) = \sum_{\mathbf{i}} \rho_{t|\mathbf{i}}(x, v)\lambda_{\mathbf{i}}. \tag{19}$$

To approximate the optimal multi-marginal transport plan $\lambda_{\mathbf{i}}$, we use the following upper bound.

**Theorem 4.** *Let $J_{\mathbf{i}}$ be the optimal cost of solving the $\mathbf{i}$-GMSB subproblem of the form ([10]) with marginal distributions $(i_1, \ldots, i_M)$ components of the marginal mixtures. Then, the cost function of the linear optimization problem*

$$\min_{\lambda_{\mathbf{i}} \geq 0} \quad J_{\text{OT}} \triangleq \sum_{\mathbf{i}} \lambda_{\mathbf{i}} J_{\mathbf{i}} \tag{20a}$$

$$\text{s.t.} \quad \sum_{\{\mathbf{i}|i_j=k\}} \lambda_{\mathbf{i}} = \alpha_j^k, \quad \forall \ j = 1, \ldots, M, \ k = 1, \ldots, N_j \tag{20b}$$

*provides an upper bound for ([16a]), that is, $J_{\text{GMM}} \leq J_{\text{OT}}$, for all values of $\lambda_{\mathbf{i}}$ satisfying ([20b]).*

In practice, in order to use policy ([18]) for inference, the initial conditions for the SDE ([16b]) must be defined. Specifically, to fully define the initial conditions in the state space $[x_0, v_0]$, given an initial position sample $x_0$, the corresponding velocity $v_0$ must be estimated. To this end, note that given $\lambda_{\mathbf{i}}$, the state distribution $\rho_t(x_t, v_t)$ is a fully defined GMM given by Equation ([19]). It is easy to show that the conditional distribution $\rho_t(v_t|x_t)$ is also a GMM, whose parameters can be easily computed. Since this calculation is trivial, we defer it to Appendix [C.3]. For completeness, we provide the multi-marginal inference algorithm in Algorithm [3]. The multi-marginal training algorithm is omitted due to its similarity to ([2]).

---

**Algorithm 3** Multi-marginal GMMflow inference

    **Input:** Component-level solutions $u_{t|\mathbf{i}}, \rho_{t|\mathbf{i}}$, multi-marginal transport plan $\lambda_{\mathbf{i}}$, sample from $x_0 \sim \rho_0(x_0)$, SDE integrator `sde_int()`.
  $u_t(x) \leftarrow$ Compute ([18])
  Sample $v_0 \sim \rho_0(v_0|x_0)$ using Equation ([B.19d])
  $[x_t; v_t] \leftarrow$ `sde_int`(Equation ([16b]), $[x_0; v_0], t \in [0, 1]$)
  **return** $x_t$

---

### 3.3 Continuous Gaussian Mixtures

The results of Section [3.1] can be extended to problems with continuous GMM boundary distributions. Specifically, we consider a bridge of the form

$$\min_{u \in \mathcal{U}} \quad \mathbb{E}\left[ \int_0^1 \|u_t(x_t)\|^2 \mathrm{d}t \right], \tag{21a}$$

$$\text{s.t.} \quad \mathrm{d}x_t = u_t(x_t)\,\mathrm{d}t, \tag{21b}$$

$$x_i \sim \int_{\mathbb{R}^m} \mathcal{N}(\mu_i(w_i), \Sigma_i(w_i))\,\mathrm{d}P_i(w_i), \quad i = 0, 1, \tag{21c}$$

where the boundary distributions ([21c]) are continuous GMMs with mixing measures $P(w_0)$, $P(w_1)$ respectively. We keep the dynamics ([21b]) deterministic to simplify the analysis, although all the results carry over to the general case of stochastic dynamics as well. This more general formulation specializes to problem ([12]) when $P(w_0)$, $P(w_1)$ have discrete support; however, it includes many other scenarios where the mixing distributions are continuous. Specifically, the generalization of Theorem [1] is as follows.

**Theorem 5.** *Consider Problem ([21]) and assume that $u_{t|w_0,w_1}$ is the conditional policy that solves the $(w_0, w_1)$-GSB problem, that is the bridge from the initial Gaussian distribution with parameter $w_0$ to the terminal Gaussian distribution with parameter $w_1$ and let the resulting probability flow be $\rho_{t|w_0,w_1}$. Furthermore, let $\Lambda(w_0, w_1)$ be any coupling such that its marginal distributions are $P_0$ and $P_1$ respectively. Then, the policy*

$$u_t(x) = \int_{\mathbb{R}^m \times \mathbb{R}^m} u_{t|w_0,w_1}(x) \frac{\rho_{t|w_0,w_1}(x)\,\mathrm{d}\Lambda(w_0, w_1)}{\int_{\mathbb{R}^m \times \mathbb{R}^m} \rho_{t|w_0,w_1}(x)\,\mathrm{d}\Lambda(w_0, w_1)} \tag{22}$$

*is a feasible policy for Problem ([21]) and the corresponding probability flow is*

$$\rho_t(x) = \int_{\mathbb{R}^m \times \mathbb{R}^m} \rho_{t|w_0,w_1}(x)\,\mathrm{d}\Lambda(w_0, w_1). \tag{23}$$

Furthermore, Theorem [2] generalizes to the following.

**Theorem 6.** *Let $J(w_0, w_1)$ be the optimal cost of solving the $(w_0, w_1)$-Gaussian bridge subproblem. Then, the optimal transport problem:*

$$J_{\mathrm{OT}} \triangleq \min_{\Lambda \in \Pi(P_0, P_1)} \int_{\mathbb{R}^m \times \mathbb{R}^m} J(w_0, w_1) \, \mathrm{d}\Lambda(w_0, w_1), \tag{24}$$

*provides an upper bound for Problem (21), that is, $J_{\mathrm{GMM}} \leq J_{\mathrm{OT}}$, where $\Pi(P_0, P_1)$ represents the set of all couplings with marginals $P_0$ and $P_1$.*

Problem (24) is challenging to solve in general. However, in many practical cases, such as for Student-t boundary distributions, the parameter spaces for $w_0, w_1$ are one-dimensional and (24) can be solved in closed form. We explore this interesting direction in Appendix B to approximate the Wasserstein-2 distance and the displacement interpolation between heavy-tail distributions.

## 4 Related Work

Although the idea of creating a mixture policy from elementary point-to-point policies is at the heart of flow-matching, to the best of our knowledge, its benefits for solving problems with mixture models with a finite number of components have not been explored. In an early work, Chen et al. (2016) developed an upper bound on the 2-Wasserstein distance between Gaussian mixture models, i.e., the static, deterministic version of Problem (12), that matches the upper bound of Theorem 2. However, the problem of finding a policy that solves the dynamic problem was not explored. Focusing on works concerning dynamic problems, the concept of constructing stochastic differential equations (SDEs) as mixtures of Gaussian probability flows can be traced back to mathematical finance applications (Brigo, 2002; Brigo et al., 2002). More recently, and in the context of generative applications, Albergo & Vanden-Eijnden (2023) developed similar expressions for finding a policy for steering between mixture models using an alternative conditional solution for the Gaussian-to-Gaussian bridge, focused on the case of deterministic, fully observable dynamical systems with no prior dynamics, and the component-level transport plan was not optimized.

Our work reveals some similarities in scope and structure with LSB (Korotin et al., 2024) and LSBM (Gushchin et al., 2024). Similarly to LSB and LSBM, we aim to provide numerically inexpensive tools to solve the SBP in low-to-moderate dimensional scenarios. Moreover, our feedback policy in Equation (13) is a mixture of affine feedback terms weighted with exponential kernels; this is also the case in LSB and LSBM (Korotin et al., 2024, Proposition 3.3). The formulations, however, are otherwise quite distinct. Specifically, LSB/LSBM works by modeling the so-called Schrödinger *potentials* using a GMM. This results in flows with boundary distributions that have a mixture-like structure and are optimal by construction, but whose marginals are neither amenable to exact calculation, nor are GMMs, in general. Finally, to approximate the optimal flows for boundary distributions available only through samples, both works solve non-linear optimization problems, which are prone to converging to locally optimal solutions. In contrast, our approach works by first pre-fitting GMMs to the data using the Expectation Maximization (EM) (Pedregosa et al., 2011), and then computing an optimal policy by solving exactly a linear program. Finally, being based on a control-theoretic framework, our method generalizes effectively to partially observable and multi-marginal distribution matching problems as shown in Section 3.2, while extending LSB and LSBM to handle such problems is non-trivial.

## 5 Experiments

**2D Problems and Benchmarks.** We first test the algorithm in various 2D "toy" problems as shown, for example, in Figure 1 for a Gaussian-to-Gaussian Mixture problem for various noise levels. To assess optimality, we evaluate the resulting transport cost for policy (13) for each noise level and compare it with the upper bound from (15a). We also run a series of EOT benchmarks and compare them with state-of-the-art neural approaches such as the DSB (De Bortoli et al., 2021) and DSBM (Shi et al., 2023) algorithms. Due to space considerations, we defer their discussion to Appendix C, along with further experiments studying training and inference time scaling with respect to the problem dimension and the number of GMM components. To evaluate the performance on problems with many GMM components, we tested the algorithm on the distributions depicted in Figure 2, where we first pre-fit 500-component GMMs in the initial and terminal samples.

**Image-to-Image Translation.** Following Korotin et al. (2024), we use our algorithm in the latent space of an autoencoder to perform a man-to-woman and adult-to-child image translation task. We

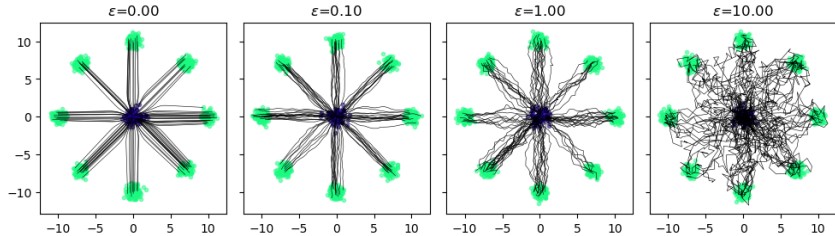

Figure 1: Gaussian to 8-Gaussians with no prior dynamics and for various levels of noise.

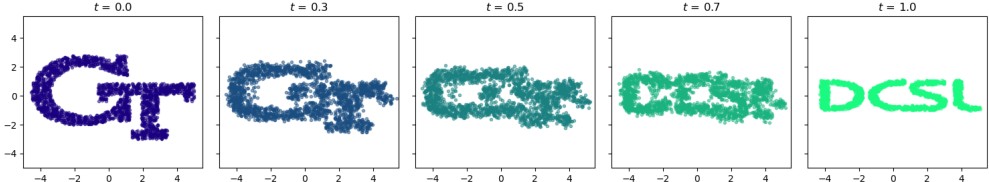

Figure 2: GT to DCSL Schrödinger Bridge with zero prior dynamics and $\epsilon = 0.1$.

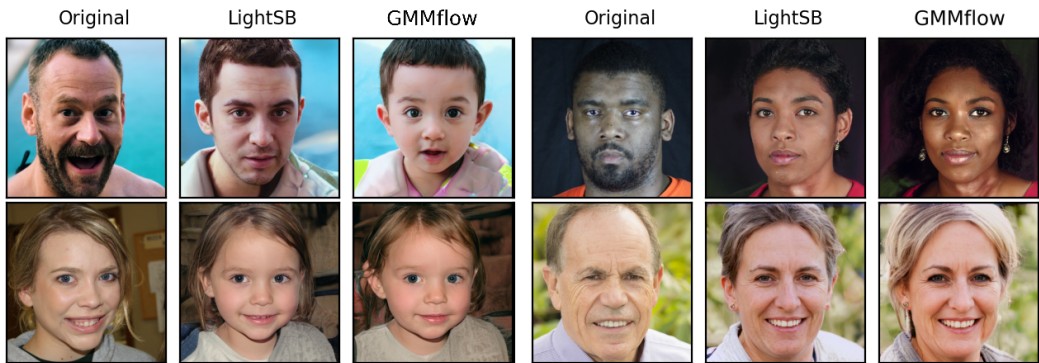

Figure 3: Adult to Child (left) and Man to Woman (right) image translation task.

use the pre-trained ALAE autoencoder (Pidhorskyi et al., 2020), trained on the FFHQ dataset (Karras et al., 2021). The latent space of the autoencoder is 512-dimensional. We start by fitting a 10-component mixture model to the embeddings of each image class, with diagonal covariance matrices to facilitate matrix inversions in the Gaussian-to-Gaussian policy calculations summarized in Proposition 1, and then apply Algorithms 1, 2 for $\epsilon = 0.01$. The results are illustrated in Figure 3.

To test how well the generated images match the features of the given target distribution, we calculate the Fréchet inception distance (FID) scores (Heusel et al., 2017) between the actual and the generated images of a given class, using 10,000 samples from each distribution. The FID scores correspond to the empirical Bures-Wasserstein distance between the images of the two classes, evaluated in the latent space of the Inception network. To further test how close the transformed images are to the target class, we also calculate the empirical Bures-Wasserstein distance between the transformed images and the real images of the target class, directly in the latent space of the ALAE autoencoder and report it as ALAE-B$\mathbb{W}$. We compare against two state-of-the-art lightweight SB solvers, namely, LSB and LSBM, with results shown in Tables 1 and 2.

Table 1: Man-to-Woman FID comparison

| M→W | FID | ALAE-B$\mathbb{W}$ | T. Cost |
|---|---|---|---|
| LSB | 4.94 | 28.9 | 8.23 |
| LSBM | 4.98 | 28.3 | 8.18 |
| **GMMflow** | 3.04 | 9.3 | 9.05 |

Table 2: Adult-to-Child FID comparison

| A→C | FID | ALAE-B$\mathbb{W}$ | T. Cost |
|---|---|---|---|
| LSB | 6.62 | 31.00 | 8.18 |
| LSBM | 6.61 | 30.99 | 8.19 |
| **GMMflow** | 3.50 | 8.54 | 9.33 |

Qualitatively, our algorithm performs more aggressive feature changes compared to the baseline method, as illustrated in Figure 3. Quantitatively, the features of the transformed images better

capture the true distribution of the features of a given target class, given the almost 40% better FID scores and 65% better ALAE-B$\mathbb{W}$ scores provided in Tables 1 and 2. We note that the improvement in the FID and ALAE-B$\mathbb{W}$ scores comes with a slight increase in the average transport cost, which we measure by $\sqrt{\mathbb{E}_\pi \|x_0 - x_1\|^2}$, as reported in the last columns of Tables 1 and 2. We attribute this significant performance gain to our use of the EM algorithm for the GMM pre-fitting, which is less prone to converge to locally optimal values, compared to LightSB's maximum likelihood objective, or the LSBM's bridge matching objective. We further remark that our approach takes 63% less time to train, as noted in Table 5 in Appendix C.2.

**Multi-Marginal Problems.** A key challenge in SBs is learning a system's underlying diffusion process, given samples from partial observations of the distribution of its state, measured at regular time intervals (Chen et al., 2023). This challenge arises, for example, in learning the dynamics of large cell populations throughout their different developmental stages (Bunne et al., 2022; Tong et al., 2020; Terpin et al., 2024b). To showcase the effectiveness of our approach in such problems, we consider the scRNA-seq dataset from (Moon et al., 2019), with the pre-processing detailed in Tong et al. (2020). The dataset contains samples from the first 100 Principal Components (PC) of individual cell proteins, grouped at 5 regular time intervals, denoted by $t_1, \ldots, t_5$.

For our setup, we keep the first 5 PCs from the 5 marginal distributions, and prefit 5-component GMMs in each marginal. We use the second-order model (16b) to capture the prior dynamics and the structure of the system; however, we note that any LTV model with structure of the form (11) would be applicable. To solve the resulting multi-marginal momentum SBs, we first compute the cost of each GMSB and then solve (20). Computing the GMSB cost for all the combinations of components is the most computationally expensive part of our approach. By parallelizing this computation, the total training time is approximately 8 minutes on an i7-12700 CPU.

We visualize the data generated by our method in Figure 4, and provide standard performance metrics in Table 3. Specifically, following Chen et al. (2023), we use the Sliced Wasserstein Distance (SWD) and Maximum Mean Discrepancy (MMD) metrics averaged over the 4 predicted time marginals of the dataset. Although there are no light-weight solvers for second-order multi-marginal problems, we compare our method against DMSB (Chen et al., 2023), NSBL (Koshizuka & Sato, 2023), and MIOFlow (Tong et al., 2020). Even though our method requires minutes to compute on a CPU while neural methods take an hour to train on a high-end GPU, the proposed approach outperforms the baselines by one order of magnitude in the MMD metric due to the accurate fitting of the GMMs. The SWD metric, although acceptable, is bounded by the expressivity of the GMMs to capture the higher-order distribution structure. The metrics for MIOFlow, NSBL, and DMSB in Table 3 are taken from (Chen et al., 2023).

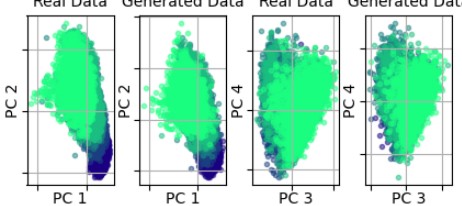

Figure 4: GMM momentum multimagrinal SB: Real vs Generated data.

Table 3: SWD and MMD indices averaged over the 4 time steps

| Method | SWD | MMD |
|---|---|---|
| MIOFlow | 0.38 | 0.28 |
| NSBL | 0.24 | 0.10 |
| DMSB | 0.22 | 0.06 |
| **GMMflow** | 0.37 $\pm$4E-3 | 0.0038 $\pm$7E-4 |

## 6    Conclusion and Limitations

This paper introduces a novel, efficient method for solving SB problems with GMM boundary distributions by utilizing a mixture of conditional policies, each solving a Gaussian bridge subproblem. In the same way that low-dimensional methods such as Gaussian distributions and mixture models are highly used in statistics and machine learning, we believe that our approach will be a valuable tool in many useful practical problems related to optimal transport, distribution interpolation, distributional control, and related applications, given its very low computational complexity, and excellent empirical performance in complicated statistical problems. The main limitation of our work stems from its dependence on GMM marginal distributions. Since this class of distributions is not designed for use in very high-dimensional problems, we do not expect our method to be applicable in such areas, but rather to work as an efficient tool for obtaining rapid SB solutions to smaller problems.

## Acknowledgments

The authors would like to thank Peter Garud for his help with the coding of the multimarginal SB implementation. Support for this work has been provided by ONR award N00014-18-1-2828 and NASA ULI award #80NSSC20M0163. This article solely reflects the opinions and conclusions of its authors and not of any NASA entity. George Rapakoulias acknowledges financial support from the A. Onassis Foundation Scholarship. Lingjiong Zhu is partially supported by NSF grants DMS-2053454 and DMS-2208303.

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

# Go With the Flow: Fast Diffusion for Gaussian Mixture Models

## Supplementary Material

The supplementary document is organized as follows.

- In Appendix A, we provide the technical proofs of the results in the main paper, an optimality analysis of Theorem 2, and the complexity analysis of Algorithm 1.

- In Appendix B, we discuss continuous mixtures and applications to heavy-tailed distributions.

- In Appendix C, we provide additional numerical experiments and some details for the experiments of Section 5 in the main paper.

- In Appendix D, we provide technical background for the Gaussian Schrödinger Bridges with LTV prior dynamics.

## A Proofs

All proofs are carried out for a stochastic Linear Time-Varying (LTV) system

$$\mathrm{d}x_t = A_t x_t \, \mathrm{d}t + B_t u(x_t) \, \mathrm{d}t + D_t \, \mathrm{d}w, \tag{A.1}$$

where $x_t \in \mathbb{R}^d$, $A_t \in \mathbb{R}^{d \times d}$, $u_t \in \mathbb{R}^m$, $B_t \in \mathbb{R}^{d \times m}$, $D_t \in \mathbb{R}^{d \times q}$ and $\mathrm{d}w$ is the $q$-dimensional Brownian increment having the properties $\mathbb{E}[\mathrm{d}w] = \mathbb{E}[\mathrm{d}w \, \mathrm{d}t] = 0$ and $\mathbb{E}[\mathrm{d}w \, \mathrm{d}w^\mathsf{T}] = I \mathrm{d}t$. The dynamical system (12b) is just a special case of (A.1) with $A_t = 0, B_t = I, D_t = \sqrt{\epsilon}\, I$, while the second order model (16b) can be captured by

$$A_t = \begin{bmatrix} 0 & I \\ 0 & 0 \end{bmatrix}, \quad B_t = \begin{bmatrix} 0 \\ I \end{bmatrix}, \quad D_t = \sqrt{\epsilon} \begin{bmatrix} 0 \\ I \end{bmatrix}. \tag{A.2}$$

### A.1 The Fokker-Planck-Kolmogorov Equation

The equation describing the propagation of the distribution of the state of the dynamical system (A.1), known as the Fokker-Planck-Kolmogorov (FPK) equation (Särkkä & Solin, 2019) is:

$$\frac{\partial \rho_t}{\partial t} + \sum_i \frac{\partial}{\partial x_i} \Big( \rho_t (A_t x + B_t u_t(x)) \Big) - \frac{1}{2} \sum_{i,j} \frac{\partial^2}{\partial x_i \partial x_j} \left( [D_t D_t^\mathsf{T}]_{ij} \rho_t \right) = 0, \tag{A.3}$$

where, for simplicity, we write $\rho_t = \rho(t, x)$. This equation can be written more compactly using standard vector notation as follows

$$\frac{\partial \rho_t}{\partial t} + \nabla \cdot \Big( \rho_t \left( A_t x + B_t u_t(x) \right) \Big) - \frac{1}{2} \mathrm{tr} \left( D_t D_t^\mathsf{T} \nabla^2 \rho_t \right) = 0, \tag{A.4}$$

where $\nabla^2 \rho_t$ denotes the Hessian of the density with respect to the state $x$ at time $t$. In the specific case where $D_t = \sqrt{\epsilon}\, I$, equation (A.4) reduces to the well-known equation

$$\frac{\partial \rho_t}{\partial t} + \nabla \cdot \Big( \rho_t (A_t x + B_t u_t(x)) \Big) - \frac{\epsilon}{2} \Delta \rho_t = 0, \tag{A.5}$$

where $\Delta$ denotes the Laplacian operator.

### A.2 Proof of Proposition 2

Let $\rho_0 \sim \mathcal{N}(\mu_0, \Sigma_0)$, $\rho_1 \sim \mathcal{N}(\mu_1, \Sigma_1)$ be two multivariate Gaussian measures in $\mathbb{R}^d$, and consider the entropy regularized 2-Wasserstein distance problem between $\rho_0$ and $\rho_1$, defined by

$$\mathbb{W}_2^\epsilon(\rho_0 \| \rho_1) \triangleq \min_{\pi \in \Pi(\rho_0, \rho_1)} \int \|x - y\|^2 \mathrm{d}\pi(x, y) + \epsilon \mathrm{D}_{\mathrm{KL}}(\pi \| \rho_0 \otimes \rho_1), \tag{A.6}$$

where $\mathrm{D}_{\mathrm{KL}}$ denotes the KL-divergence operator.

In (Mallasto et al., 2022, Theorem 2), it is shown that (A.6) admits the following closed-form solution

$$\mathbb{W}_2^\epsilon(\rho_0 \| \rho_1) = \|\mu_1 - \mu_0\|^2 + \mathrm{tr}(\Sigma_0) + \mathrm{tr}(\Sigma_1) - \frac{\epsilon}{2} \left( \mathrm{tr} M_\epsilon - \log \det M_\epsilon + d \log 2 - 2d \right), \tag{A.7}$$

where $M_\epsilon = I + (I + (4/\epsilon)^2 \Sigma_0 \Sigma_1)^{\frac{1}{2}}$.

Returning to Problem (6), let $\mathcal{P}, \mathcal{W}^\epsilon$ be the path measures corresponding to the controlled SDE of (6) (expressed as the first constraint) and the Brownian motion with covariance $\epsilon I$ respectively, with initial conditions sampled from $\rho_0$. Let $\mathcal{D}(\rho_0, \rho_1)$ denote the set of all path measures with marginals $\rho_0, \rho_1$ at times $t = 0$ and $t = 1$. Then, the SB problem (6) admits the following equivalent representations:

$$\inf_{\mathcal{P} \in \mathcal{D}(\rho_0, \rho_1)} D_{KL}(\mathcal{P} \| \mathcal{W}^\epsilon) = \inf_{\pi \in \Pi(\rho_0, \rho_1)} \left\{ \int \frac{\|x-y\|^2}{2\epsilon} d\pi(x,y) - H(\pi) + H(\rho_0) + \frac{d}{2} \log(2\pi e) \right\} \tag{A.8a}$$

$$= \inf_{\mathcal{P} \in \mathcal{D}(\rho_0, \rho_1)} \mathbb{E} \left[ \int_0^1 \frac{1}{2\epsilon} \|u_t(x_t)\|^2 dt \right] = \frac{1}{2\epsilon} J_{GSB}, \tag{A.8b}$$

where (A.8a) is due to the disintegration of the path measures, and (A.8b) is due to Girsanov's Theorem. We refer the reader to Chen et al. (2021) for a detailed derivation. Solving (A.8b) for $J_{GSB}$, we obtain

$$J_{GSB} = \inf_\pi \left\{ \int \|x - y\|^2 d\pi(x,y) - 2\epsilon H(\pi) - 2\epsilon H(\rho_1) \right\}, \tag{A.9}$$

up to a constant independent of the parameters of $\rho_0, \rho_1$. Using the closed form solution for (A.7), and noting that

$$D_{KL}(\pi \| \rho_0 \otimes \rho_1) = -H(\pi) + H(\rho_0) + H(\rho_1), \tag{A.10}$$

and that the differential entropy of the multivariate Gaussian distribution is

$$H(\rho_1) = \frac{1}{2} \log \det \Sigma_1 + \frac{d}{2} \log 2\pi e, \tag{A.11}$$

we conclude that the optimal cost of Problem (6) is equal to

$$\begin{aligned} J_{GSB} &= \mathbb{W}_2^{2\epsilon}(\rho_0 \| \rho_1) - 2\epsilon H(\rho_1) \\ &= \|\mu_1 - \mu_0\|^2 + \text{tr}(\Sigma_0) + \text{tr}(\Sigma_1) - \epsilon \left( \text{tr} M_{2\epsilon} - \log \det M_{2\epsilon} + \log \det \Sigma_1 \right), \end{aligned} \tag{A.12a}$$

up to a numerical constant independent of $\rho_0, \rho_1$, which concludes the proof.

### A.3 Proof of Theorem 1

First, notice that the probability flow (14) satisfies the constraints (12c) for all feasible values of $\lambda_{ij}$, since

$$\rho_0 = \sum_{i,j} \rho_{0|ij} \lambda_{ij} = \sum_{i,j} \mathcal{N}(\mu_0^i, \Sigma_0^i) \lambda_{ij} = \sum_i \mathcal{N}(\mu_0^i, \Sigma_0^i) \alpha_0^i, \tag{A.13a}$$

$$\rho_1 = \sum_{i,j} \rho_{1|ij} \lambda_{ij} = \sum_{i,j} \mathcal{N}(\mu_1^j, \Sigma_1^i) \lambda_{ij} = \sum_j \mathcal{N}(\mu_1^j, \Sigma_1^i) \alpha_1^j. \tag{A.13b}$$

Therefore, it suffices to show that the policy (13) produces the probability flow (14). Following the approach of Lipman et al. (2023) and Liu et al. (2024), we show that the pair (13), (14) satisfies the FPK equation. We start from the FPK equation describing a conditional flow and sum over all conditional variables to retrieve the unconditional flow. Specifically, given that the individual policies $u_{t|ij}$ solve the Gaussian Bridge subproblems (6), the pair $(\rho_{t|ij}, u_{t|ij})$ satisfies the FPK equation for the dynamical system (A.1), that is,

$$\frac{\partial \rho_{t|ij}}{\partial t} + \nabla \cdot \left( \rho_{t|ij} \left( A_t x + B_t u_{t|ij} \right) \right) - \frac{1}{2} \text{tr} \left( D_t D_t^\mathsf{T} \nabla^2 (\rho_{t|ij}) \right) = 0. \tag{A.14}$$

Multiplying equation (A.14) by $\lambda_{ij}$ and summing over $i, j$, we obtain

$$\sum_{i,j} \lambda_{ij} \left[ \frac{\partial \rho_{t|ij}}{\partial t} + \nabla \cdot \left( \rho_{t|ij} \left( A_t x + B_t u_{t|ij} \right) \right) - \frac{1}{2} \text{tr} \left( D_t D_t^\mathsf{T} \nabla^2 (\rho_{t|ij}) \right) \right] = 0, \tag{A.15}$$

which implies that

$$\frac{\partial}{\partial t}\left(\sum_{i,j}\rho_{t|ij}\lambda_{ij}\right) + \nabla \cdot \left(A_t x \sum_{i,j}\rho_{t|ij}\lambda_{ij} + B_t \sum_{i,j}u_{t|ij}\rho_{t|ij}\lambda_{ij}\right)$$

$$-\frac{1}{2}\mathrm{tr}\left(D_t D_t^\mathsf{T}\nabla^2\left(\sum_{i,j}\rho_{t|ij}\lambda_{ij}\right)\right) = 0. \qquad (A.16)$$

This can be further simplified as

$$\frac{\partial \rho_t}{\partial t} + \nabla \cdot \left(\rho_t\left(A_t x + B_t \sum_{ij}u_{t|ij}\frac{\rho_{t|ij}\lambda_{ij}}{\sum_{i,j}\rho_{t|ij}\lambda_{ij}}\right)\right) - \frac{1}{2}\mathrm{tr}\left(D_t D_t^\mathsf{T}\nabla^2(\rho_t)\right) = 0, \qquad (A.17)$$

which yields that

$$\frac{\partial \rho_t}{\partial t} + \nabla \cdot \left(\rho_t(A_t x + B_t u_t)\right) - \frac{1}{2}\mathrm{tr}\left(D_t D_t^\mathsf{T}\nabla^2(\rho_t)\right) = 0. \qquad (A.18)$$

This completes the proof.

### A.4  Proof of Theorem 2

It suffices to show that the cost (12a) is upper bounded by the cost of (15a). Substituting policy (13) to the cost (12a) we obtain

$$J_{\mathrm{GMM}} = \mathbb{E}_{x_t \sim \rho_t}\left[\int_0^1 \left\|\sum_{i,j}u_{t|ij}(x_t)\frac{\rho_{t|ij}(x_t)\lambda_{ij}}{\sum_{i,j}\rho_{t|ij}(x_t)\lambda_{ij}}\right\|^2 \mathrm{d}t\right] \qquad (A.19a)$$

$$= \int_0^1 \int \rho_t(x)\left\|\sum_{i,j}u_{t|ij}(x)\frac{\rho_{t|ij}(x)\lambda_{ij}}{\sum_{i,j}\rho_{t|ij}(x)\lambda_{ij}}\right\|^2 \mathrm{d}x\,\mathrm{d}t \qquad (A.19b)$$

$$\leq \int_0^1 \int \rho_t(x)\frac{\sum_{i,j}\left\|u_{t|ij}(x)\right\|^2 \rho_{t|ij}(x)\lambda_{ij}}{\sum_{i,j}\rho_{t|ij}(x)\lambda_{ij}}\,\mathrm{d}x\,\mathrm{d}t \qquad (A.19c)$$

$$= \int_0^1 \int \sum_{i,j}\left\|u_{t|ij}(x)\right\|^2 \rho_{t|ij}(x)\lambda_{ij}\,\mathrm{d}x\,\mathrm{d}t \qquad (A.19d)$$

$$= \sum_{i,j}\lambda_{ij}\mathbb{E}_{x_t \sim \rho_{t|ij}}\left[\int_0^1 \left\|u_{t|ij}(x_t)\right\|^2 \mathrm{d}t\right] \qquad (A.19e)$$

$$= \sum_{i,j}\lambda_{ij}J_{ij} = J_{\mathrm{OT}}, \qquad (A.19f)$$

where (A.19b) is due to Fubini's theorem (Wheeden & Zygmund, 1977, Theorem 6.1) and (A.19c) makes use of the discrete version of Jensen's inequality (Wheeden & Zygmund, 1977, Theorem 7.35).

### A.5  Optimality of the Upper Bound of Theorem 2.

To assess the optimality of the upper bound introduced in Theorem 2, we study the gap between $J_{\mathrm{OT}}$ and $J_{\mathrm{GMM}}$ in the following theorem.

**Theorem 7.** *In the setting of Theorem 2, let $\rho_{t|ij}(x), u_{t|ij}(x)$ be the solution of the $(i,j)$-GSB and $u_t(x), \rho_t(x)$ as defined in (13), (14). Then, the following bound holds for $J_{\mathrm{OT}}, J_{\mathrm{GMM}}$.*

$$0 \leq J_{\mathrm{OT}} - J_{\mathrm{GMM}} \leq \int_0^1 \int_{\mathbb{R}^n}\sum_{i,j}\sum_{\substack{i'\neq i\\ j'\neq j}}\left\|u_{t|ij} - u_{t|i'j'}\right\|^2 \min\{\lambda_{ij}\rho_{t|ij}, \lambda_{i'j'}\rho_{t|i'j'}\}\,\mathrm{d}x\,\mathrm{d}t, \qquad (A.20)$$

*where the dependence on $x$ is omitted for notational convenience.*

*Proof.* We first express $u_t(x)$ as an expectation, i.e.,

$$u_t(x) = \sum_{i,j} u_{t|ij}(x) \frac{\rho_{t|ij}(x)\lambda_{ij}}{\rho_t(x)} = \mathbb{E}[\omega_t(x)],$$

where $\omega_t(x)$ follows a discrete distribution defined by $\{\omega_t(x) = u_{t|ij}(x)$ w.p. $\rho_{t|ij}(x)\lambda_{ij}/\rho_t(x)\}$. Note that for a random variable $x \in \mathbb{R}^d$, the variance decomposition yields

$$\|\mathbb{E}[x]\|^2 = \mathbb{E}[\|x\|^2] - \mathbb{E}[\|x - \mathbb{E}[x]\|^2].$$

Using the last equation and the expression $u_t(x)$ above, written as an expectation, we obtain

$$
\begin{aligned}
J_{\mathrm{GMM}} &\triangleq \int_0^1 \int_{\mathbb{R}^d} \rho_t(x) \|u_t(x)\|^2 \, \mathrm{d}x \, \mathrm{d}t \\
&= \int_0^1 \int_{\mathbb{R}^d} \sum_{i,j} \lambda_{ij} \rho_{t|i,j} \|u_{t|i,j}(x)\|^2 \, \mathrm{d}x \, \mathrm{d}t - \int_0^1 \int_{\mathbb{R}^d} \sum_{i,j} \lambda_{ij} \rho_{t|i,j} \|u_{t|i,j}(x) - u_t(x)\|^2 \, \mathrm{d}x \, \mathrm{d}t \\
&= J_{\mathrm{OT}} - \int_0^1 \int_{\mathbb{R}^d} \sum_{i,j} \lambda_{ij} \rho_{t|ij} \|u_{t|ij} - u_t\|^2 \, \mathrm{d}x \, \mathrm{d}t.
\end{aligned}
$$

The fact that the second term in the last equation is non-negative justifies the upper bound in Theorem 2. Next, we show that when the conditional densities are well separated, the second term in the last equation becomes arbitrarily small, and the bound becomes tight. Expanding the term inside the norm in the integral of the last equation, and dropping the dependence on $x$ for notational convenience, we obtain

$$
\begin{aligned}
\|u_{t|i,j} - u_t\|^2 &= \left\| u_{t|i,j} - \sum_{i'j'} u_{t|i'j'} \frac{\lambda_{i'j'} \rho_{t|i'j'}}{\rho_t} \right\|^2 \\
&= \left\| \frac{u_{t|ij} \rho_t - \sum_{i'j'} u_{t|i'j'} \lambda_{i'j'} \rho_{t|i'j'}}{\rho_t} \right\|^2 \\
&\leq \left( \sum_{i'j'} \|u_{t|i,j} - u_{t|i'j'}\| \frac{\lambda_{i'j'} \rho_{t|i'j'}}{\rho_t} \right)^2 \quad \text{(due to Jensen's inequality)} \\
&\leq \sum_{i'j'} \|u_{t|i,j} - u_{t|i'j'}\|^2 \frac{\lambda_{i'j'} \rho_{t|i'j'}}{\rho_t} \\
&= \sum_{\substack{i' \neq i \\ j' \neq j}} \|u_{t|i,j} - u_{t|i'j'}\|^2 \frac{\lambda_{i'j'} \rho_{t|i'j'}}{\rho_t}.
\end{aligned}
$$

Substituting this upper bound in the expression for $J_{\mathrm{GMM}}$ and rearranging, we get

$$
\begin{aligned}
0 &\leq J_{\mathrm{OT}} - J_{\mathrm{GMM}} \\
&\leq \int_0^1 \int_{\mathbb{R}^d} \sum_{i,j} \lambda_{ij} \rho_{t|i,j} \|u_{t|i,j}(x) - u_t(x)\|^2 \, \mathrm{d}x \, \mathrm{d}t \\
&\leq \int_0^1 \int_{\mathbb{R}^d} \sum_{i,j} \sum_{\substack{i' \neq i \\ j' \neq j}} \|u_{t|ij} - u_{t|i'j'}\|^2 \frac{\lambda_{i'j'} \lambda_{ij} \rho_{t|i'j'} \rho_{t|i,j}}{\rho_t} \, \mathrm{d}x \, \mathrm{d}t \\
&\leq \int_0^1 \int_{\mathbb{R}^d} \sum_{i,j} \sum_{\substack{i' \neq i \\ j' \neq j}} \|u_{t|ij} - u_{t|i'j'}\|^2 \min\{\lambda_{ij} \rho_{t|ij}, \lambda_{i'j'} \rho_{t|i'j'}\} \, \mathrm{d}x \, \mathrm{d}t,
\end{aligned}
$$

where the last inequality comes from the inequality $\frac{a_i a_j}{\sum_i a_i} \leq \min(a_i, a_j)$ for all positive numbers $\{a_i\}_{i=1}^N$. This completes the proof. $\qquad\square$

Letting

$$Q = \frac{\int \min\{\lambda_{ij}\rho_{t|ij}, \lambda_{i'j'}\rho_{t|i'j'}\}\, dt}{\iint \min\{\lambda_{ij}\rho_{t|ij}, \lambda_{i'j'}\rho_{t|i'j'}\}\, dx\, dt},$$

that is, the normalized distribution of the minimum of the densities $\rho_{t|ij}, \rho_{t|i'j'}$, and assuming that $\mathbb{E}_Q[\|u_{t|ij}(x) - u_{t|i',j'}(x)\|^2] < \infty$, since the policies $u_{t|ij}$ are affine with respect to $x$, we conclude that

$$J_{\mathrm{OT}} - J_{\mathrm{GMM}} \to 0 \quad \text{as} \quad \mathrm{TV}(\rho_{t|ij}, \rho_{t|i'j'}) \to 1 \quad \forall i, j, i', j', (i,j) \neq (i', j'), \tag{A.21}$$

where $\mathrm{TV}(\mu, \nu)$ denotes the total variation between two probability measures.

## A.6 Training and Inference complexity of GMMflow

In this section, we provide a computational complexity analysis of Algorithm 1 with respect to the number of components in each mixture and the problem dimension. The computational complexity of fitting a GMM using the EM algorithm scales as $O(INK(D + D^2))$ (Pedregosa et al., 2011), where $I$ is the number of EM iterations, $N$ is the number of data points, $K$ is the number of Gaussian components (modes), and $D$ is the dimensionality of the data. Once the GMMs are fitted, solving a linear program with $N_0 \times N_1$ variables, where $N_0$ and $N_1$ denote the number of modes in the input and output distributions, respectively. Modern solvers such as MOSEK (Mosek, 2020) efficiently solve LP problems using interior-point methods, which have a computational complexity of $O(\sqrt{l}N^3)$ (Boyd & Vandenberghe, 2004), where $l$ represents the number of constraints.

Regarding the computational complexity of inference, each evaluation of the GMMflow policy, i.e., equation (13), scales linearly with the number of components in each mixture and the SDE integration also scales linearly with the number of discretization time steps. In practice, when implementing the GMMflow policy, only a small number of GSB policies are computed since the component level transport plan $\lambda_{ij}$ is sparse. Moreover, this computation is done in parallel for all conditional policies together. This results in very fast, practically constant-time inference regardless of the component number or problem dimension.

## A.7 Proof of Theorem 3

We start by noting that we can write the dynamical system (16b) in the form of (A.1) with

$$A_t = \begin{bmatrix} 0 & I \\ 0 & 0 \end{bmatrix}, \quad B_t = \begin{bmatrix} 0 \\ I \end{bmatrix}, \quad D_t = \begin{bmatrix} 0 \\ \sqrt{\epsilon}I \end{bmatrix}. \tag{A.22}$$

Due to (19) the joint density of the phase space $(x_t, v_t)$ is given by

$$\rho_t(x, v) = \sum_{\mathbf{i}} \lambda_{\mathbf{i}} \mathcal{N}\left(\begin{bmatrix} x \\ v \end{bmatrix}; \begin{bmatrix} \mu_{t|\mathbf{i}}^x \\ \mu_{t|\mathbf{i}}^v \end{bmatrix}, \begin{bmatrix} \Sigma_{t|\mathbf{i}}^{xx} & \Sigma_{t|\mathbf{i}}^{xv} \\ \Sigma_{t|\mathbf{i}}^{vx} & \Sigma_{t|\mathbf{i}}^{vv} \end{bmatrix}\right), \tag{A.23}$$

We now note that for all $j = 1, \ldots, M$, the position marginal $x_{t_j}$ at time $t_j$ is distributed as

$$x_{t_j} \sim \sum_{\mathbf{i}} \lambda_{\mathbf{i}} \mathcal{N}(\mu_{t_j|\mathbf{i}}^x, \Sigma_{t_j|\mathbf{i}}^{xx}) = \sum_{k=1}^{N_j} \sum_{\{\mathbf{i}: i_j = k\}} \lambda_{\mathbf{i}} \mathcal{N}(\mu_j^k, \Sigma_j^k) = \sum_{k=1}^{N_j} \alpha_j^k \mathcal{N}(\mu_j^k, \Sigma_j^k), \tag{A.24}$$

and therefore the flow (19) satisfies the constraint (16c). Using a similar approach to the proof of Theorem 1, we will show that (18) produces the probability flow (19) by summing over all conditional GMSB flows. To facilitate notation, we will denote the phase space by $z \in \mathbb{R}^{2d}$, i.e., $z = [x; v]$. Given that the individual policies $u_{t|\mathbf{i}}$ solve the GMSB subproblems (10), the pair $(\rho_{t|\mathbf{i}}, u_{t|\mathbf{i}})$ satisfies the FPK equation for the dynamical system (A.1), that is,

$$\frac{\partial \rho_{t|\mathbf{i}}}{\partial t} + \nabla \cdot \left(\rho_{t|\mathbf{i}}\left(Az + Bu_{t|\mathbf{i}}\right)\right) - \frac{1}{2}\mathrm{tr}\left(DD^\mathsf{T}\nabla^2(\rho_{t|\mathbf{i}})\right) = 0. \tag{A.25}$$

Multiplying equation (A.25) by $\lambda_{\mathbf{i}}$ and summing over $\mathbf{i}$, we obtain

$$\sum_{\mathbf{i}} \lambda_{\mathbf{i}} \left[\frac{\partial \rho_{t|\mathbf{i}}}{\partial t} + \nabla \cdot \left(\rho_{t|\mathbf{i}}\left(Az + Bu_{t|\mathbf{i}}\right)\right) - \frac{1}{2}\mathrm{tr}\left(DD^\mathsf{T}\nabla^2(\rho_{t|\mathbf{i}})\right)\right] = 0, \tag{A.26}$$

which implies that

$$\frac{\partial}{\partial t}\left(\sum_{\mathbf{i}}\rho_{t|\mathbf{i}}\lambda_{\mathbf{i}}\right) + \nabla \cdot \left(Az\sum_{\mathbf{i}}\rho_{t|\mathbf{i}}\lambda_{\mathbf{i}} + B\sum_{\mathbf{i}}u_{t|\mathbf{i}}\rho_{t|\mathbf{i}}\lambda_{\mathbf{i}}\right)$$
$$-\frac{1}{2}\mathrm{tr}\left(DD^{\mathsf{T}}\nabla^2\left(\sum_{\mathbf{i}}\rho_{t|\mathbf{i}}\lambda_{\mathbf{i}}\right)\right) = 0. \tag{A.27}$$

This can be further simplified as

$$\frac{\partial \rho_t}{\partial t} + \nabla \cdot \left(\rho_t\left(Az + B\sum_{\mathbf{i}}u_{t|\mathbf{i}}\frac{\rho_{t|\mathbf{i}}\lambda_{\mathbf{i}}}{\sum_{\mathbf{i}}\rho_{t|\mathbf{i}}\lambda_{\mathbf{i}}}\right)\right) - \frac{1}{2}\mathrm{tr}\left(DD^{\mathsf{T}}\nabla^2(\rho_t)\right) = 0, \tag{A.28}$$

which yields that

$$\frac{\partial \rho_t}{\partial t} + \nabla \cdot \left(\rho_t(Az + Bu_t)\right) - \frac{1}{2}\mathrm{tr}\left(DD^{\mathsf{T}}\nabla^2(\rho_t)\right) = 0. \tag{A.29}$$

This completes the proof.

## A.8 Proof of Theorem 4

The proof is similar to that of Theorem 2. Substituting policy (18) to the cost (16a) we obtain

$$J_{\mathrm{GMM}} = \mathbb{E}_{z_t \sim \rho_t}\left[\int_0^1 \left\|\sum_{\mathbf{i}}u_{t|\mathbf{i}}(z_t)\frac{\rho_{t|\mathbf{i}}(z_t)\lambda_{\mathbf{i}}}{\sum_{\mathbf{i}}\rho_{t|\mathbf{i}}(z_t)\lambda_{\mathbf{i}}}\right\|^2 \mathrm{d}t\right] \tag{A.30a}$$

$$= \int_0^1 \int \rho_t(z)\left\|\sum_{\mathbf{i}}u_{t|\mathbf{i}}(z)\frac{\rho_{t|\mathbf{i}}(z)\lambda_{\mathbf{i}}}{\sum_{\mathbf{i}}\rho_{t|\mathbf{i}}(z)\lambda_{\mathbf{i}}}\right\|^2 \mathrm{d}z\,\mathrm{d}t \tag{A.30b}$$

$$\leq \int_0^1 \int \rho_t(z)\frac{\sum_{\mathbf{i}}\left\|u_{t|\mathbf{i}}(z)\right\|^2\rho_{t|\mathbf{i}}(z)\lambda_{\mathbf{i}}}{\sum_{\mathbf{i}}\rho_{t|\mathbf{i}}(z)\lambda_{\mathbf{i}}}\,\mathrm{d}z\,\mathrm{d}t \tag{A.30c}$$

$$= \int_0^1 \int \sum_{\mathbf{i}}\left\|u_{t|\mathbf{i}}(z)\right\|^2\rho_{t|\mathbf{i}}(z)\lambda_{\mathbf{i}}\,\mathrm{d}z\,\mathrm{d}t \tag{A.30d}$$

$$= \sum_{\mathbf{i}}\lambda_{\mathbf{i}}\mathbb{E}_{z_t \sim \rho_{t|\mathbf{i}}}\left[\int_0^1 \left\|u_{t|\mathbf{i}}(z_t)\right\|^2 \mathrm{d}t\right] \tag{A.30e}$$

$$= \sum_{\mathbf{i}}\lambda_{\mathbf{i}}J_{\mathbf{i}} = J_{\mathrm{OT}}, \tag{A.30f}$$

where (A.30b) is due to Fubini's theorem (Wheeden & Zygmund, 1977, Theorem 6.1) and (A.30c) makes use of the discrete version of Jensen's inequality (Wheeden & Zygmund, 1977, Theorem 7.35).

## A.9 Proof of Theorem 5

The proof is similar to the (discrete) GMM case. First, notice that

$$\rho_0 = \int_{\mathbb{R}^m \times \mathbb{R}^m}\rho_{0|w_0,w_1}\mathrm{d}\Lambda(w_0,w_1) = \int_{\mathbb{R}^m}\mathcal{N}(\mu_0(w_0),\Sigma_0(w_0))\mathrm{d}P_0(w_0), \tag{A.31}$$

$$\rho_1 = \int_{\mathbb{R}^m \times \mathbb{R}^m}\rho_{1|w_0,w_1}\mathrm{d}\Lambda(w_0,w_1) = \int_{\mathbb{R}^m}\mathcal{N}(\mu_1(w_1),\Sigma_1(w_1))\mathrm{d}P_1(w_1). \tag{A.32}$$

Next, notice that $\rho_{t|w_0,w_1}$ and $u_{t|w_0,w_1}$ satisfy the FPK equation:

$$\frac{\partial \rho_{t|w_0,w_1}}{\partial t} + \nabla \cdot \left(\rho_{t|w_0,w_1}(A_tx_t + B_tu_{t|w_0,w_1})\right) - \frac{1}{2}\mathrm{tr}\left(D_tD_t^{\mathsf{T}}\nabla^2(\rho_{t|w_0,w_1})\right) = 0. \tag{A.33}$$

By taking the expectation with respect to the distribution $\Lambda(w_0,w_1)$ in (A.33), we get

$$\int_{\mathbb{R}^m \times \mathbb{R}^m}\left[\frac{\partial \rho_{t|w_0,w_1}}{\partial t} + \nabla \cdot \left(\rho_{t|w_0,w_1}(A_tx_t + B_tu_{t|w_0,w_1})\right)\right.$$
$$\left. - \frac{1}{2}\mathrm{tr}\left(D_tD_t^{\mathsf{T}}\nabla^2(\rho_{t|w_0,w_1})\right)\right]\mathrm{d}\Lambda(w_0,w_1) = 0, \tag{A.34}$$

which implies that

$$\frac{\partial}{\partial t}\int_{\mathbb{R}^m\times\mathbb{R}^m}\rho_{t|w_0,w_1}\mathrm{d}\Lambda(w_0,w_1)$$

$$+\nabla\cdot\left(A_t x_t\int_{\mathbb{R}^m\times\mathbb{R}^m}\rho_{t|w_0,w_1}\mathrm{d}\Lambda(w_0,w_1)+B_t\int_{\mathbb{R}^m\times\mathbb{R}^m}u_{t|w_0,w_1}\rho_{t|w_0,w_1}\mathrm{d}\Lambda(w_0,w_1)\right)$$

$$-\frac{1}{2}\mathrm{tr}\left(D_t D_t^\mathsf{T}\nabla^2\left(\int_{\mathbb{R}^m\times\mathbb{R}^m}\rho_{t|w_0,w_1}\mathrm{d}\Lambda(w_0,w_1)\right)\right)=0, \tag{A.35}$$

which yields that

$$\frac{\partial\rho_t}{\partial t}+\nabla\cdot\left(\rho_t\left(A_t x_t+B_t\int_{\mathbb{R}^m\times\mathbb{R}^m}u_{t|w_0,w_1}\frac{\rho_{t|w_0,w_1}\mathrm{d}\Lambda(w_0,w_1)}{\int_{\mathbb{R}^m\times\mathbb{R}^m}\rho_{t|w_0,w_1}\mathrm{d}\Lambda(w_0,w_1)}\right)\right)$$

$$-\frac{1}{2}\mathrm{tr}\left(D_t D_t^\mathsf{T}\nabla^2(\rho_t)\right)=0. \tag{A.36}$$

Hence, we conclude that

$$\frac{\partial\rho_t}{\partial t}+\nabla\cdot(\rho_t(A_t x_t+B_t u_t))-\frac{1}{2}\mathrm{tr}\left(D_t D_t^\mathsf{T}\nabla^2(\rho_t)\right)=0. \tag{A.37}$$

### A.10 Proof of Theorem 6

The proof is similar to the (discrete) GMM case. We can compute that

$$J_{\mathrm{GMM}}=\mathbb{E}_{x_t\sim\rho_t}\left[\int_0^1\left\|\int_{\mathbb{R}^m\times\mathbb{R}^m}u_{t|w_0,w_1}(x)\frac{\rho_{t|w_0,w_1}(x)\mathrm{d}\Lambda(w_0,w_1)}{\int_{\mathbb{R}^m\times\mathbb{R}^m}\rho_{t|w_0,w_1}(x)\mathrm{d}\Lambda(w_0,w_1)}\right\|^2\mathrm{d}t\right] \tag{A.38}$$

$$=\int_0^1\int_{\mathbb{R}^n}\rho_t\left\|\int_{\mathbb{R}^m\times\mathbb{R}^m}u_{t|w_0,w_1}(x)\frac{\rho_{t|w_0,w_1}(x)\mathrm{d}\Lambda(w_0,w_1)}{\int_{\mathbb{R}^m\times\mathbb{R}^m}\rho_{t|w_0,w_1}(x)\mathrm{d}\Lambda(w_0,w_1)}\right\|^2\mathrm{d}x\mathrm{d}t \tag{A.39}$$

$$\leq\int_0^1\int_{\mathbb{R}^n}\rho_t\frac{\int_{\mathbb{R}^m\times\mathbb{R}^m}\|u_{t|w_0,w_1}(x)\|^2\rho_{t|w_0,w_1}(x)\mathrm{d}\Lambda(w_0,w_1)}{\int_{\mathbb{R}^m\times\mathbb{R}^m}\rho_{t|w_0,w_1}(x)\mathrm{d}\Lambda(w_0,w_1)}\mathrm{d}x\mathrm{d}t \tag{A.40}$$

$$=\int_0^1\int_{\mathbb{R}^n}\int_{\mathbb{R}^m\times\mathbb{R}^m}\|u_{t|w_0,w_1}(x)\|^2\rho_{t|w_0,w_1}(x)\mathrm{d}\Lambda(w_0,w_1)\mathrm{d}x\mathrm{d}t \tag{A.41}$$

$$=\int_{\mathbb{R}^m\times\mathbb{R}^m}\mathbb{E}_{x\sim\rho_{t|w_0,w_1}}\left[\int_0^1\|u_{t|w_0,w_1}(x)\|^2\mathrm{d}t\right]\mathrm{d}\Lambda(w_0,w_1). \tag{A.42}$$

Hence, for any $\Lambda\in\Pi(P_0,P_1)$,

$$J_{\mathrm{GMM}}\leq\int_{\mathbb{R}^m\times\mathbb{R}^m}J(w_0,w_1)\mathrm{d}\Lambda(w_0,w_1). \tag{A.43}$$

By taking the infimum over $\Lambda\in\Pi(P_0,P_1)$ in (A.43), we conclude that $J_{\mathrm{GMM}}\leq J_{\mathrm{OT}}$.

## B  Continuous Gaussian Mixtures

Theorem 6 reduces the high-dimensional dynamic optimal transport problem (21) to the simpler, static OT problem (24) in the space of couplings between the parameter distributions, i.e., $\Pi(P_0,P_1)$. Although in general, problem (24) is still difficult to solve, in many practical applications the parameter spaces $w_0,w_1$ are one-dimensional and $J(w_0,w_1)$ has a tractable closed form. Under these conditions, and provided the distributions $P_0,P_1$ admit positive densities $p_0,p_1$, (24) can be solved in almost closed form (Santambrogio, 2015).

Our motivation for the extension to continuous mixtures stems from the fact that many heavy-tail distributions, such as the multivariate Student-t distribution and the alpha-stable distribution, can be expressed in the form of continuous Gaussian mixtures. In this context, one can use a generalized version of Theorems 1 and 2 to create tractable upper bounds on the 2-Wasserstein distance between such distributions and approximate the corresponding optimal transport map, displacement interpolation, and flow fields, respectively.

## B.1 Multivariate $t$-Distribution

The Student-t distribution has been used as a heavy-tailed alternative to the Gaussian distribution, as a generative prior distribution in diffusion models and related generative models in the recent literature; see e.g., Kim et al. (2024); Pandey et al. (2025); Cordero-Encinar et al. (2025). In this section, we will explore the use cases of Theorems 5 and 6 to the case of Student-t boundary distributions.

To this end, let $x_0, x_1$ follow $d$-dimensional multivariate $t$-distributions with parameters $\nu_0, \mu_0, \Sigma_0$ and $\nu_1, \mu_1, \Sigma_1$ respectively.[2]

A multivariate $t$-distribution can be viewed as a generalized Gaussian mixture model; see, for example, Andrews & Mallows (1974). More specifically, let $u_0, u_1$ follow a gamma distribution,[3] i.e.,

$$u_0 \sim \mathrm{Gamma}(\nu_0/2, \nu_0/2), \qquad u_1 \sim \mathrm{Gamma}(\nu_1/2, \nu_1/2), \tag{B.2}$$

and conditional on $u_0, u_1$, $x_0 \sim \mathcal{N}(\mu_0, u_0^{-1}\Sigma_0)$ and $x_1 \sim \mathcal{N}(\mu_1, u_1^{-1}\Sigma_1)$ respectively. Then, $w_0^2 = u_0^{-1}$ follows the distribution $\mathrm{InverseGamma}(\nu_0/2, \nu_0/2)$ and $w_1^2 = u_1^{-1}$ follows the distribution $\mathrm{InverseGamma}(\nu_1/2, \nu_1/2)$, that is, $w_0^2$ has the probability density function

$$\frac{(2/\nu_0)^{\nu_0/2}}{\Gamma(\nu_0/2)}(1/x)^{\frac{\nu_0}{2}+1}e^{-\frac{2}{\nu_0 x}}, \tag{B.3}$$

and $w_1^2$ has the probability density function

$$\frac{(2/\nu_1)^{\nu_1/2}}{\Gamma(\nu_1/2)}(1/x)^{\frac{\nu_1}{2}+1}e^{-\frac{2}{\nu_1 x}}. \tag{B.4}$$

Let $P_0, P_1, p_0, p_1$ the CDFs and PDFs of $w_0, w_1$ respectively. Starting with the CDF of $w_0$, we have

$$P_0(x) = \mathbb{P}(w_0 \leq x) = \mathbb{P}(w_0^2 \leq x^2) = \int_0^{x^2} \frac{(2/\nu_0)^{\nu_0/2}}{\Gamma(\nu_0/2)}(1/y)^{\frac{\nu_0}{2}+1}e^{-\frac{2}{\nu_0 y}}\,\mathrm{d}y = \frac{\Gamma(\frac{\nu_0}{2}, \frac{\nu_0}{2x^2})}{\Gamma(\frac{\nu_0}{2}, 0)}, \tag{B.5}$$

which implies that $w_0 \sim P_0$ has the probability density function

$$p_0(x) = \frac{\mathrm{d}}{\mathrm{d}x}\mathbb{P}(w_0 \leq x) = 2x\frac{(2/\nu_0)^{\nu_0/2}}{\Gamma(\nu_0/2)}(1/x^2)^{\frac{\nu_0}{2}+1}e^{-\frac{2}{\nu_0 x^2}} = \frac{2(2/\nu_0)^{\nu_0/2}}{\Gamma(\nu_0/2)}(1/x)^{\nu_0+1}e^{-\frac{2}{\nu_0 x^2}}. \tag{B.6}$$

Similarly, $w_1 \sim P_1$ has a CDF given by

$$P_1(x) = \frac{\Gamma(\frac{\nu_1}{2}, \frac{\nu_1}{2x^2})}{\Gamma(\frac{\nu_1}{2}, 0)}, \tag{B.7}$$

and a PDF given by

$$p_1(x) = \frac{2(2/\nu_1)^{\nu_1/2}}{\Gamma(\nu_1/2)}(1/x)^{\nu_1+1}e^{-\frac{2}{\nu_1 x^2}}. \tag{B.8}$$

With equations (B.5)-(B.8) in mind, consider Problem (21) with Student-t distributions with parameters $\nu_0, \mu_0, \Sigma_0$ and $\nu_1, \mu_1, \Sigma_1$, that is, continuous mixtures of the form

$$\rho_0(x) = \int \mathcal{N}(x; \mu_0, w_0^2\Sigma_0)p_0(w_0)\mathrm{d}w_0, \tag{B.9a}$$

$$\rho_1(x) = \int \mathcal{N}(x; \mu_1, w_1^2\Sigma_1)p_1(w_1)\mathrm{d}w_1, \tag{B.9b}$$

---

[2]The density of a multivariate t-distribution with $\nu$ degrees of freedom, and scale and location parameters $\mu, \Sigma$ respectively, is given by

$$\frac{\Gamma((\nu+d)/2)}{\Gamma(\nu/2)\nu^{d/2}\pi^{d/2}|\Sigma|^{1/2}}\left[1 + \frac{1}{\nu}(x-\mu)^\top\Sigma^{-1}(x-\mu)\right]^{-(\nu+d)/2}. \tag{B.1}$$

[3]Here $\mathrm{Gamma}(a, b)$ denotes a gamma distribution with probability density functional proportional to $x^{a-1}e^{-bx}$ where $a$ is the shape parameter and $b$ is the inverse scale parameter.

where $p_0(w_0), p_1(w_1)$ are given by (B.6) and (B.8) respectively.

Considering noise-free dynamics to simplify the respective formulas, as in (21b), the $(w_0\text{-}w_1)$-GSB for the boundary distribution parametrization (B.9), admits the following closed form

$$\Sigma_{t|w_0,w_1} = (1-t)^2 w_0^2 \Sigma_0 + t^2 w_1^2 \Sigma_1 + (1-t)t w_0 w_1 \left( C + C^\mathsf{T} \right), \tag{B.10a}$$

$$\mu_{t|w_0,w_1} = (1-t)\mu_0 + t\mu_1, \tag{B.10b}$$

$$K_{t|w_0,w_1} = S_t^\mathsf{T} \Sigma_t^{-1}, \tag{B.10c}$$

$$v_{t|w_0,w_1} = \mu_1 - \mu_0, \tag{B.10d}$$

$$S_{t|w_0,w_1} = t \left( \Sigma_1 - C^\mathsf{T} \right) - (1-t)(\Sigma_0 - C), \tag{B.10e}$$

where $C = \Sigma_0^{\frac{1}{2}} D \Sigma_0^{-\frac{1}{2}}$, $D = (\Sigma_0^{\frac{1}{2}} \Sigma_1 \Sigma_0^{\frac{1}{2}})^{\frac{1}{2}}$. Equations (B.10) yield

$$\rho_{t|w_0,w_1}(x) = \mathcal{N}\left(x; \mu_{t|w_0 w_1}, \Sigma_{t|w_0,w_1}\right), \tag{B.11a}$$

$$u_{t|w_0,w_1}(x) = K_{t|w_0,w_1}\left(x - \mu_{t|w_0,w_1}\right) + \mu_1 - \mu_0. \tag{B.11b}$$

Furthermore, the optimal cost $J(w_0, w_1)$ in (24) can be calculated using (9) and equals

$$J(w_0, w_1) = \|\mu_0 - \mu_1\|^2 + w_0^2 \mathrm{tr}(\Sigma_0) + w_1^2 \mathrm{tr}(\Sigma_1) - 2w_0 w_1 \mathrm{tr}(D). \tag{B.12}$$

Focusing on problem (24), it is known that when the transport cost has the form $J(w_0, w_1) = h(w_0 - w_1)$ where $h$ is a convex function, the corresponding optimal transport plan is given by (Santambrogio, 2015, Theorem 2.9)

$$\Lambda^*(w_0, w_1) = \left( P_0^{-1}(x), P_1^{-1}(x) \right)_\# \mathrm{Unif}([0,1]), \tag{B.13}$$

where $\mathrm{Unif}([0,1])$ is the uniform measure over the set $[0,1]$, and $P_0^{-1}, P_1^{-1}$ are the corresponding inverse CDFs of (B.5), (B.7). The cost function (B.12), does not satisfy this condition, since it cannot be written as a perfect square for general scaling matrices $\Sigma_0, \Sigma_1$. We resolve this issue through the following proposition.

**Proposition 3.** *When solving the OT problem (24), the transport cost (B.12) and the cost function $\tilde{J}(w_0, w_1) = |w_0 - w_1|^2$ are equivalent, i.e., they result in the same optimal coupling $\Lambda^*$.*

*Proof.* Equation (B.12) can be written in the form:

$$J(w_0, w_1) = \|\mu_0 - \mu_1\|^2 + w_0^2 \mathrm{tr}(\Sigma_0 - D) + w_1^2 \mathrm{tr}(\Sigma_1 - D) + |w_0 - w_1|^2 \mathrm{tr}(D). \tag{B.14}$$

In (B.14), the terms $\|\mu_0 - \mu_1\|^2$, $w_0^2 \mathrm{tr}(\Sigma_0 - D)$, $w_1^2 \mathrm{tr}(\Sigma_1 - D)$ do not contribute to the optimization problem in (24), since they are constant for any feasible transport plan $\Lambda \in \Pi(\rho_0, \rho_1)$ due to the fixed boundary distributions. By dropping these terms, as well as the positive scaling constant $\mathrm{tr}(D)$, we obtain the desired result. $\square$

Equation (B.13) implies that the optimal value for Problem (24) is given by

$$J_{\mathrm{OT}}^* = \int_0^1 J\left(P_0^{-1}(w), P_1^{-1}(w)\right) \mathrm{d}w, \tag{B.15}$$

while the optimal control policy $u_t^*(x)$ and the respective density $\rho_t^*(x)$ resulting from substituting to the optimal transport plan $\Lambda^*$ to the formulas (22) and (5) of Theorem 5 are given by

$$\rho_t^*(x) = \int_0^1 \rho_{t|P_0^{-1}(w),P_1^{-1}(w)}(x)\mathrm{d}w, \tag{B.16a}$$

and

$$u_t^*(x) = \int_0^1 u_{t|P_0^{-1}(w),P_1^{-1}(w)}(x) \frac{\rho_{t|P_0^{-1}(w),P_1^{-1}(w)}(x)}{\rho_t^*(x)} \mathrm{d}w. \tag{B.17}$$

Since equations (B.15)-(B.17) involve only one-dimensional integrals, they can be easily computed numerically using a quadrature. Although $P_0^{-1}, P_1^{-1}$ are not available in closed form, they can be

obtained in most scientific computing packages such as `scipy` (Virtanen et al., 2020) by properly scaling the quantile function of the inverse gamma distribution.

To illustrate this approach, we calculate the upper bound (B.15) and the true Wasserstein-2 distance for a one-dimensional problem between two Student-t distributions with parameters $\mu_1 = \mu_2 = 0$, $\Sigma_1 = 1$, $\Sigma_2 = \{0.25, 1, 4\}$, $\nu_1 = 3$ for various values of $\nu_2 \in [2.5, 10]$ and report the results in Figure 5. We note that our approach works for arbitrary Student-t distributions in any dimension, however, we study the 1D case in this example to be able to calculate the exact Wasserstein distance and quantify the tightness of our upper bound. Although rigorously studying the tightness of the bound (B.15) remains an open problem, as evident in Figure 5, it closely approximates the true Wasserstein distance, at-least in this simple 1D scenario.

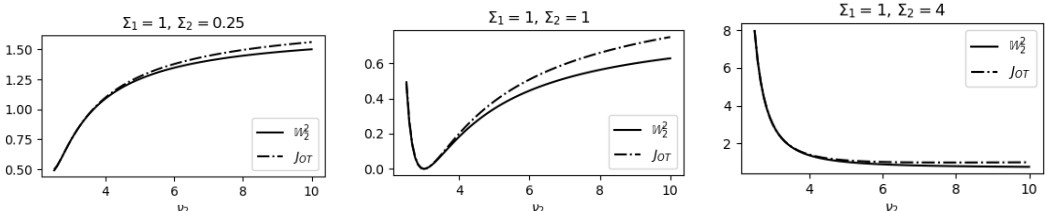

Figure 5: Comparison between true Wasserstein distance and upper bound (B.15) for 1D Student-t distributions.

We believe this result, along with the policy (B.17) and the interpolation (B.16), could be useful tools in developing simulation-free methods for training diffusion models with heavy-tail prior distributions, or for creating tractable OT flows between mixtures of Student-t distributions. We leave these interesting directions as future work, since they are not immediately related to computationally inexpensive diffusion model training, which is the main theme of the rest of this paper.

# C  Additional Experiments and Implementation Details

## C.1  Additional Details on 2D Problems

To compare our approach for the problem of Figure 1 with state-of-the-art neural SB solvers we used the original implementations of the DSB[4] (De Bortoli et al., 2021) and DSBM[5] (Shi et al., 2023). The network architecture used for both algorithms is the fully connected DNN of De Bortoli et al. (2021) with 128-dimensional sinusoidal temporal encodings, 256 neurons in the encoder layer, $\{256, 256\}$ neurons in the decoder layers, and SiLU activation functions (Hendrycks & Gimpel, 2016). We run all algorithms and report the results in Table 4. For the zero noise case, i.e., $\epsilon = 0$, DSB and DSBM are not applicable, so we approximate the true OT cost using discrete optimal transport, calculated using the POT library (Flamary et al., 2021), using 10,000 samples from each distribution. As evident from Table 4, although the DSB and DSBM algorithms can approximate the true SB cost, they fail to retrieve the true optimal solution for larger values of the noise parameter, due to their non-convex loss functions.

Table 4: Transport cost comparison for the problem in Figure 1

| $\epsilon$ | $J_{\mathrm{OT}}$ (12a) | $J_{\mathrm{GMM}}$ (15a) | DSBM | DSB | OT |
|---|---|---|---|---|---|
| 0 | 100.06 | 89.45 | - | - | 84.87 |
| 0.1 | 100.15 | 89.32 | 84.26 | 98.62 | - |
| 1 | 102.28 | 89.28 | 131.50 | 100.82 | - |
| 10 | 162.13 | 116.60 | 133.04 | 244.31 | - |

Furthermore, regarding the example problem in Figure 2, we provide additional details about the approximation of the boundary distributions as mixture models in Figure 6.

## C.2  Image-to-Image Translation Details

To better evaluate the performance of the proposed approach in the Image-to-Image translation task, we provide further examples in Figures 9 and 10 as well as approximate training and inference times

---

[4] https://github.com/JTT94/diffusion_schrodinger_bridge
[5] https://github.com/yuyang-shi/dsbm-pytorch

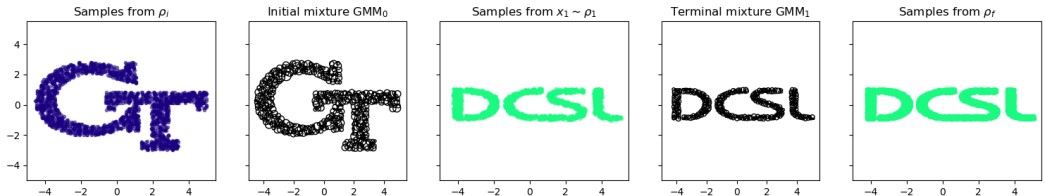

Figure 6: GT to DCSL distribution steering details.

for our approach, and compare them with the training and inference times of LightSB in Table 5. For our approach, training time consists of the time required to fit the GMMs in the latents of the FFHQ dataset for the two boundary distributions, and the solution of the linear program (15). As inference time, we consider the time taken for the integration of the SDE (or ODE for $\epsilon = 0$) (12b) with the mixture policy (13). We observe that while inference time is small for solving the deterministic (optimal transport) problem, i.e., for $\epsilon = 0$, integrating the stochastic dynamical system for positive values of $\epsilon$ requires more time due to the small time step required for SDE integration. The quality of the produced images was not found to be affected by this parameter, implying that $\epsilon = 0$ could be used for fast, deterministic inference, while a positive value of $\epsilon$ will allow for some randomness in the generated images. We also note that the faster training time for our approach is mainly due to the very fast convergence of the EM algorithm, which is also less likely to converge to local minima, compared to the standard maximum likelihood method for fitting distribution to data. All tests were conducted on a desktop computer with an RTX 3070 GPU.

Table 5: Training and inference time comparison with state of the art. Inference time is measured for a batch of 10 images and uses GPU parallelization for calculating the elementary GSB policies of the mixture policy (13).

|          | Training [s] | Inference ($\epsilon = 0$) [s], | Inference ($\epsilon = 0.1$) [s] |
|----------|--------------|---------------------------------|----------------------------------|
| LightSB[6] | 57         | -                               | 0.02                             |
| Ours     | 17           | 0.06                            | 0.2                              |

## C.3 Multi-Marginal Problems Details

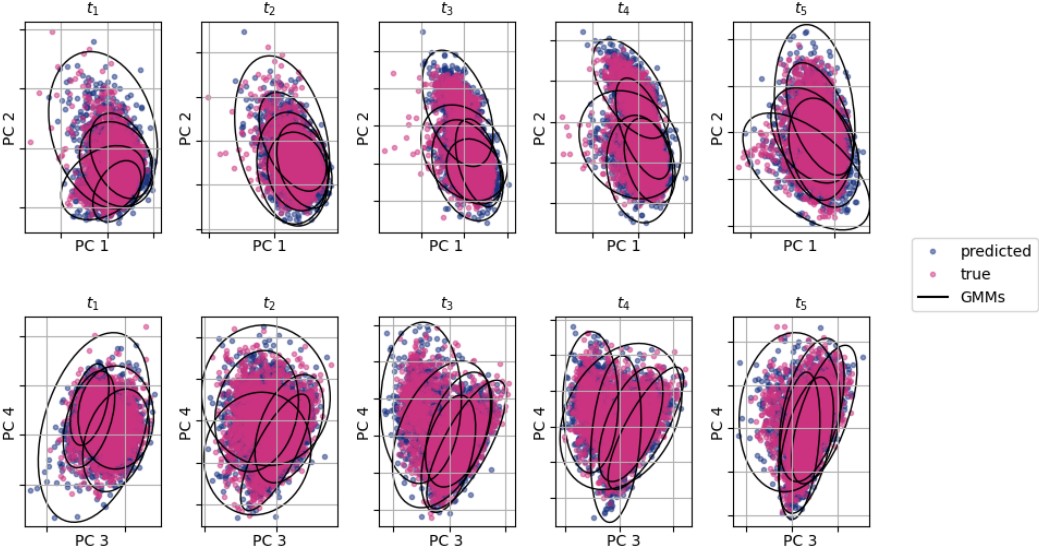

Figure 7: Additional visualization of results in 5-d scRNA problem: predicted vs true distributions for all time-marginals overlayed with the 3-sigma bound for each Gaussian component of the pre-fitted GMMs.

---

[6]Korotin et al. (2024)

**Solution times.** To solve each multi-marginal GMSB we use the semidefinite formulation detailed in Section D and used Mosek (2020) to solve the resulting semidefinite program. Specifically, we assume the GMMs between the five temporal marginals are spaced 1 time unit apart, resulting in a problem horizon of 4 time units. We use a coarse temporal discretization with time-step $\Delta t = 0.1$ (i.e., 10 time steps between $[t_i, t_{i+1}]$) to evaluate the cost tensor for the optimization problem (20) and a fine resolution discretization of $\Delta t = 0.01$ for the final policy calculation, solving only for the GMSBs with non-zero transport parameter $\lambda_i$. There are a total of $3,125$ combinations of GMSBs for this problem, and the computation of each one using the coarse time grid takes roughly $0.35$ s on an Intel i7 12-th generation CPU with 32 GB of RAM memory, giving a total of $18.5$ minutes of calculations, if all GMSBs are solved serially. This computational overhead can be greatly decreased if the GMSBs are solved in parallel. In our setup, we parallelized the calculation using MOSEK's built-in capabilities and solved them in batches of 12, using 2 CPU threads per problem. This brought down the total calculation time under 6 minutes. For the final policy calculation, there are only 21 active GMSBs in the mixture policy (18), each taking 6 seconds to compute. The total run time for our algorithm for this problem, adds up to 8 minutes for this problem, which is considerably lower than the corresponding neural methods (24 minutes on GPU for the DMSB algorithm (Chen et al., 2023)).

**Velocity inference.** After calculating the conditional GMSBs and solving (4), the marginal mixture distribution for the entire phase-space is fully defined for the entire time horizon of the problem through Equation (19). Given a position sample $x_0$ at time $t = 0$, the corresponding velocity component can be inferred using conditional GMM sampling. Specifically, considering that the joint distribution of the phase space at time $t$ is

$$\rho_t(x, v) = \sum_{\mathbf{i}} \lambda_{\mathbf{i}} \mathcal{N}\left(\begin{bmatrix} x \\ v \end{bmatrix}; \begin{bmatrix} \mu_{t|\mathbf{i}}^x \\ \mu_{t|\mathbf{i}}^v \end{bmatrix}, \begin{bmatrix} \Sigma_{t|\mathbf{i}}^{xx} & \Sigma_{t|\mathbf{i}}^{xv} \\ \Sigma_{t|\mathbf{i}}^{vx} & \Sigma_{t|\mathbf{i}}^{vv} \end{bmatrix}\right), \tag{B.18}$$

it is easy to show that the density of $\rho_t(v|x)$ is also a Gaussian Mixture Model, since

$$\rho_t(v|x) = \frac{\rho_t(x, v)}{\rho_t(x)} \tag{B.19a}$$

$$= \frac{\sum_{\mathbf{i}} \lambda_{\mathbf{i}} \mathcal{N}\left(\begin{bmatrix} x \\ v \end{bmatrix}; \begin{bmatrix} \mu_{t|\mathbf{i}}^x \\ \mu_{t|\mathbf{i}}^v \end{bmatrix}, \begin{bmatrix} \Sigma_{t|\mathbf{i}}^{xx} & \Sigma_{t|\mathbf{i}}^{xv} \\ \Sigma_{t|\mathbf{i}}^{vx} & \Sigma_{t|\mathbf{i}}^{vv} \end{bmatrix}\right)}{\sum_{\mathbf{i}} \lambda_{\mathbf{i}} \mathcal{N}\left(x; \mu_{t|\mathbf{i}}^x, \Sigma_{t|\mathbf{i}}^{xx}\right)} \tag{B.19b}$$

$$= \frac{\sum_{\mathbf{i}} \lambda_{\mathbf{i}} \mathcal{N}\left(v; \mu_{t|\mathbf{i}}^{v|x}, \Sigma_{t|\mathbf{i}}^{v|x}\right) \mathcal{N}\left(x; \mu_{t|\mathbf{i}}^x, \Sigma_{t|\mathbf{i}}^{xx}\right)}{\sum_{\mathbf{i}} \lambda_{\mathbf{i}} \mathcal{N}\left(x; \mu_{t|\mathbf{i}}^x, \Sigma_{t|\mathbf{i}}^{xx}\right)} \tag{B.19c}$$

$$= \sum_{\mathbf{i}} \frac{\lambda_{\mathbf{i}} \mathcal{N}\left(x; \mu_{t|\mathbf{i}}^x, \Sigma_{t|\mathbf{i}}^{xx}\right)}{\sum_{\mathbf{i}} \lambda_{\mathbf{i}} \mathcal{N}\left(x; \mu_{t|\mathbf{i}}^x, \Sigma_{t|\mathbf{i}}^{xx}\right)} \mathcal{N}\left(v; \mu_{t|\mathbf{i}}^{v|x}, \Sigma_{t|\mathbf{i}}^{v|x}\right), \tag{B.19d}$$

where Equation (B.19a) is due to the Bayes rule, with (Bishop & Nasrabadi, 2006)

$$\mu_{t|\mathbf{i}}^{v|x} = \mu_{t|\mathbf{i}}^v + \Sigma_{t|\mathbf{i}}^{vx} \left(\Sigma_{t|\mathbf{i}}^{xx}\right)^{-1} \left(x - \mu_{t|\mathbf{i}}^x\right),$$

and

$$\Sigma_{t|\mathbf{i}}^{v|x} = \Sigma_{t|\mathbf{i}}^{vv} - \Sigma_{t|\mathbf{i}}^{vx} \left(\Sigma_{t|\mathbf{i}}^{xx}\right)^{-1} \Sigma_{t|\mathbf{i}}^{xv}.$$

For our problem, we use Equation (B.19d) to sample the initial velocity of a new sample given its initial position, and then use the joint position-velocity initial conditions to calculate the sample's trajectory by integrating (12b).

**Visualization of results.** To better visualize our results, we provide more information about the predicted distributions in Figure 7, overlaid with the pre-fitted GMMs at each time step. It is easy to visually confirm that the 5-component mixtures capture the marginal distributions in all time-steps accurately, and since our method is exact, there is minimal distribution mismatch in the predicted marginals, as confirmed quantitatively by the indices in Table 3.

## C.4 Performance on EOT Benchmarks

To further evaluate the optimality of the proposed approach, we tested the algorithm on the Entropic Optimal Transport benchmark detailed in Gushchin et al. (2023). The benchmark provides a pair of boundary test distributions $\rho_0, \rho_1$, where $\rho_0$ is a scaled Gaussian distribution and $\rho_1$ is a mixture-like distribution that is easy to sample from, and an optimal conditional transport plan $\pi^*(x_1|x_0)$, which is a Gaussian Mixture Model and is known in closed form. For the pair $\rho_0, \rho_1$, the optimal policy solving (2) can be calculated explicitly, allowing direct comparisons with our approach. The metric we use to measure the optimality of our approach is the Bures-Wasserstein Unexplained Variance Percentage (cBW-UVP) (Gushchin et al., 2023), defined by

$$\text{cBW-UVP}(\hat{\pi}, \pi^*) \triangleq \frac{100\%}{\frac{1}{2}\text{Var}(\rho_1)} \int \text{BW}_2^2(\hat{\pi}(x_1|x_0)\|\pi^*(x_1|x_0))\rho_0(x_0)\,\mathrm{d}x_0, \tag{B.20}$$

which measures the distance between conditional transport plans, evaluated using the Bures-Wasserstein metric.

To use the method of Gushchin et al. (2023), we first obtain samples from the two boundary test distributions and then fit mixture models on them using EM. We then deploy policy (13), and report the values of the cBW-UVP index between the known optimal conditional transport plan $\pi^*(x_1|x_0)$, and the conditional transport plan resulting from the integration of the policy (13), denoted $\hat{\pi}(x_1|x_0)$. We use 1,000 initial condition samples $x_0$, and for each sample, we draw 1,000 $x_1$ samples from the distributions $\pi^*(x_1|x_0)$ and $\hat{\pi}(x_1|x_0)$ to compute the empirical Bures-Wasserstein distance in (B.20). The results are reported in Table 6 for problems of various dimensions and noise levels, along with many other available methods for solving the same problem (Gushchin et al., 2023, Table 5). We note that although our approach requires virtually no training compared to computationally expensive neural OT and SB approaches, it outperforms many of these algorithms, outlining its excellent performance in problems where GMMs accurately capture the marginal distributions of the problem.

Table 6: Comparisons of cBW$_2^2$-UVP $\downarrow$ (%) between the optimal plan $\pi^*$ and the learned plan $\hat{\pi}$. **Colors** indicate the ratio of the metric to the *independent baseline* metric: ratio $\leq 0.2$, ratio $\in (0.2, 0.5)$, ratio $> 0.5$.

| | $\epsilon=0.1$ | | | | $\epsilon=1$ | | | | $\epsilon=10$ | | | |
|---|---|---|---|---|---|---|---|---|---|---|---|---|
| | $D=2$ | $D=16$ | $D=64$ | $D=128$ | $D=2$ | $D=16$ | $D=64$ | $D=128$ | $D=2$ | $D=16$ | $D=64$ | $D=128$ |
| ⌊LSOT⌉ | - | - | - | - | - | - | - | - | - | - | - | - |
| ⌊SCONES⌉ | - | - | - | - | 34.88 | 71.34 | 59.12 | 136.44 | 32.9 | 50.84 | 60.44 | 52.11 |
| ⌊NOT⌉ | 1.94 | 13.67 | 11.74 | 11.4 | 4.77 | 23.27 | 41.75 | 26.56 | 2.86 | 4.57 | 3.41 | 6.56 |
| ⌊EgNOT⌉ | 129.8 | 75.2 | 60.4 | 43.2 | 80.4 | 74.4 | 63.8 | 53.2 | 4.14 | 2.64 | 2.36 | 1.31 |
| ⌊ENOT⌉ | 3.64 | 22 | 13.6 | 12.6 | 1.04 | 9.4 | 21.6 | 48 | 1.4 | 2.4 | 19.6 | 30 |
| ⌊MLE-SB⌉ | 4.57 | 16.12 | 16.1 | 17.81 | 4.13 | 9.08 | 18.05 | 15.226 | 1.61 | 1.27 | 3.9 | 12.9 |
| ⌊DiffSB⌉ | 73.54 | 59.7 | 1386.4 | 1683.6 | 33.76 | 70.86 | 53.42 | 156.46 | - | - | - | - |
| ⌊FB-SDE-A⌉ | 86.4 | 53.2 | 1156.82 | 1566.44 | 30.62 | 63.48 | 34.84 | 131.72 | - | - | - | - |
| ⌊FB-SDE-J⌉ | 51.34 | 89.16 | 119.32 | 173.96 | 29.34 | 69.2 | 155.14 | 177.52 | - | - | - | - |
| ⌊DSBM⌉ | 5.2 | 16.8 | 37.3 | 35 | 0.3 | 1.1 | 9.7 | 31 | 3.7 | 105 | 3557 | 15000 |
| ⌊SF$^2$ M-Sink⌉ | 0.54 | 3.7 | 9.5 | 10.9 | 0.2 | 1.1 | 9 | 23 | 0.31 | 4.9 | 319 | 819 |
| ⌊LightSB⌉ | 0.03 | 0.08 | 0.28 | 0.60 | 0.05 | 0.09 | 0.24 | 0.62 | 0.07 | 0.11 | 0.21 | 0.37 |
| ⌊LightSB-M (MB)⌉ | 0.005 | 0.07 | 0.27 | 0.63 | 0.002 | 0.04 | 0.12 | 0.36 | 0.04 | 0.07 | 0.11 | 0.23 |
| ⌊GMMflow (ours)⌉ | 10.35 | 14.68 | 11.15 | 11.2 | 5.78 | 7.20 | 6.93 | 6.38 | 0.16 | 0.28 | 1.43 | 2.77 |
| Independent coupling | 166.0 | 152.0 | 126.0 | 110.0 | 86.0 | 80.0 | 72.0 | 60.0 | 4.2 | 2.52 | 2.26 | 2.4 |

To further benchmark our algorithm with respect to run-times, we provide wall-clock times for both training and inference with respect to the number of components and the problem dimensionality for the boundary distributions provided in the EOT benchmark. We report these values in Table 7 below.

Table 7: Training time for for EOT benchmark.

| Dim\ # comp | 5 | 10 | 20 | 50 | 100 |
|---|---|---|---|---|---|
| 2 | 0.209 | 0.0595 | 0.1083 | 0.2303 | 1.0256 |
| 16 | 0.0567 | 0.0948 | 0.1413 | 0.4218 | 3.3918 |
| 64 | 0.182 | 0.2228 | 0.7217 | 1.2875 | 2.1431 |
| 128 | 0.2802 | 0.4562 | 0.7713 | 1.6164 | 3.4101 |

Table 8: Inference time for for EOT benchmark.

| Dim\ # comp | 5 | 10 | 20 | 50 | 100 |
|---|---|---|---|---|---|
| 2 | 0.037 | 0.031 | 0.031 | 0.032 | 0.030 |
| 16 | 0.032 | 0.032 | 0.032 | 0.033 | 0.032 |
| 64 | 0.032 | 0.032 | 0.032 | 0.032 | 0.039 |
| 128 | 0.032 | 0.033 | 0.036 | 0.032 | 0.082 |

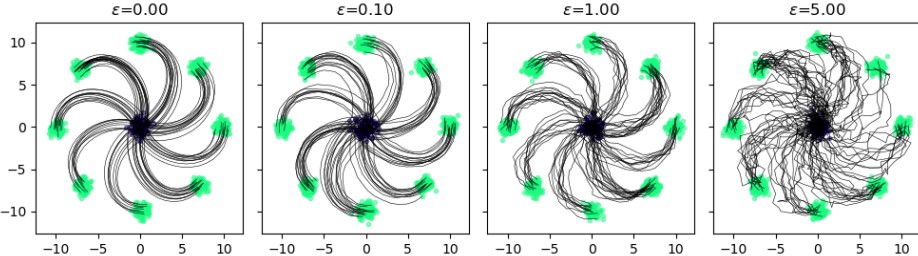

Figure 8: Gaussian to 8-Gaussians with LTI prior dynamics.

We note that because the EOT benchmark uses an initial Gaussian distribution and a GMM terminal distribution, there is no point in reporting metrics such as marginal distribution accuracy or transport plan optimality, since these will perform best when the number of components used in GMMflow matches the setting of EOT benchmark. Furthermore, exploring how well a GMM approximates a general distribution as the number of components increases is a well-studied problem and goes beyond the scope of our work; therefore, we do not provide experiments that explore this issue.

### C.5   Problems with LTI Prior Dynamics

To test the algorithm on more complicated dynamical systems, we use the 4-dimensional Linear Time-Invariant (LTI) system

$$\mathrm{d}x_t = Ax_t\,\mathrm{d}t + Bu_t\,\mathrm{d}t + D\,\mathrm{d}w, \tag{B.21}$$

with

$$A = \begin{bmatrix} 2S & I_2 \\ S & 0_2 \end{bmatrix}, \quad S = \begin{bmatrix} 0 & -1 \\ 1 & 0 \end{bmatrix}, \quad B = \begin{bmatrix} 0 \\ I_2 \end{bmatrix}, \quad D = \epsilon I_4,$$

and boundary distributions

$$\rho_0 = \sum_{k=0}^{8} \frac{1}{8}\mathcal{N}\left([10\cos(k\pi/4); 10\sin(k\pi/4); 0; 0]\,, 0.4I_4\right), \tag{B.22a}$$

$$\rho_1 = \mathcal{N}\left(0_4, 0.4I_4\right). \tag{B.22b}$$

We note that solving problem (12) with the dynamical system (B.21) in place of (12b) is not currently solvable using any mainstream neural SB solvers because the stochastic disturbance $\mathrm{d}w$ in (B.21) does not enter through the same channels as the control signal $u_t$ and the state $x_t$. The only available method to solve this problem is detailed in Chen et al. (2016), which, however, assumes access to the solution of the static EOT problem (1) with boundary distributions (B.22), and a closed form of the probability density transition kernel included by the dynamical system (B.21) for $u_t \equiv 0$. The results of our approach are illustrated in Figure 8.

To solve the Gaussian Bridge sub-problems with a dynamical system of the form (B.21) we use the discrete-time convex formulation of Rapakoulias & Tsiotras (2023). We also include a brief overview of the method in Appendix D. Each continuous-time Gaussian Bridge is discretized (in the temporal dimension) into 101 steps over uniform intervals of size $\Delta t = 0.01$. We used MOSEK (Mosek, 2020) to solve the resulting semidefinite programs.

# D   Gaussian Bridge for Linear Time-Varying Systems

In this section, we briefly review the available methods in the literature to solve the Gaussian Bridge problem with general LTV dynamics of the form (A.1). That is, we consider the problem

$$\min_{u \in \mathcal{U}} \ \mathbb{E}\left[\int_0^1 \|u_t(x)\|^2 \mathrm{d}t\right], \tag{C.1a}$$

$$\mathrm{d}x_t = A_t x_t \, \mathrm{d}t + B_t u(x_t) \, \mathrm{d}t + D_t \, \mathrm{d}w, \tag{C.1b}$$

$$x_0 \sim \mathcal{N}(\mu_0, \Sigma_0), \quad x_1 \sim \mathcal{N}(\mu_1, \Sigma_1). \tag{C.1c}$$

The solution of problem (C.1) is used to solve the Gaussian Bridge problem for the example in Section C.5 and is relevant to applications with prior dynamics of more general structure such as mean field games (Bensoussan et al., 2016) and large multi-agent control applications (Saravanos et al., 2023) or higher-order distribution interpolation problems (Chen et al., 2018, 2019). The existence and uniqueness of solutions for problem (C.1) are studied in Chen et al. (2015a); Liu et al. (2025); Liu & Tsiotras (2024). Since the state of (C.1b) remains Gaussian throughout the steering horizon, i.e., $x_t \sim \mathcal{N}(\mu_t, \Sigma_t)$, the problem simplifies to that of the control of the first two statistical moments of the state, namely the mean $\mu_t$ and the covariance $\Sigma_t$. Using a control policy parametrization of the form

$$u_t(x) = K_t(x - \mu_t) + v_t, \tag{C.2}$$

allows for the decoupling of the propagation equations for the mean and covariance of the state. More specifically, applying (C.2) to (C.1b), the equations describing the propagation of $\mu_t$ and $\Sigma_t$ yield (Särkkä & Solin, 2019, Section 5.5)

$$\dot{\Sigma}_t = (A_t + B_t K_t)\Sigma_t + \Sigma_t(A_t + B_t K_t)^\mathsf{T} + D_t D_t^\mathsf{T}, \tag{C.3a}$$

$$\dot{\mu}_t = A_t \mu_t + B_t v_t. \tag{C.3b}$$

Expanding the expression (C.3a) and performing the change of variables $U_t = K_t \Sigma_t$, we obtain

$$\dot{\Sigma}_t = A_t \Sigma_t + \Sigma_t A_t^\mathsf{T} + B_t U_t + U_t^\mathsf{T} B_t^\mathsf{T} + D_t D_t^\mathsf{T}, \tag{C.4}$$

which is linear in $U_t, \Sigma_t$. Furthermore, substituting (C.2) into the cost function (C.1a) and using the cyclic property of the trace operator along with the standard properties of the expectation yields

$$\mathbb{E}\left[\int_0^1 \|u_t(x)\|^2 \, \mathrm{d}t\right] = \int_0^1 v_t^\mathsf{T} v_t + \mathrm{tr}\left(K_t \Sigma_t K_t^\mathsf{T}\right) \, \mathrm{d}t = \int_0^1 v_t^\mathsf{T} v_t + \mathrm{tr}\left(U_t \Sigma_t^{-1} U_t^\mathsf{T}\right) \, \mathrm{d}t. \tag{C.5}$$

Equations (C.3b), (C.4), (C.5) can be used to reformulate problem (C.1) to a simpler optimization problem in the space of affine feedback policies, parameterized by $U_t$ and $v_t$. To be more precise, problem (C.1) reduces to

$$\min_{\mu_t, v_t, \Sigma_t, U_t} \ \int_0^1 v_t^\mathsf{T} v_t + \mathrm{tr}\left(U_t \Sigma_t^{-1} U_t^\mathsf{T}\right) \, \mathrm{d}t, \tag{C.6a}$$

$$\dot{\Sigma}_t = A_t \Sigma_t + \Sigma_t A_t^\mathsf{T} + B_t U_t + U_t^\mathsf{T} B_t^\mathsf{T} + D_t D_t^\mathsf{T}, \tag{C.6b}$$

$$\dot{\mu}_t = A_t \mu_t + B_t v_t, \tag{C.6c}$$

which can be further relaxed to a convex semi-definite program using the lossless convex relaxation (Chen et al., 2015b)

$$\min_{\mu_t, v_t, \Sigma_t, U_t} \ \int_0^1 v_t^\mathsf{T} v_t + \mathrm{tr}(Y_t) \, \mathrm{d}t, \tag{C.7a}$$

$$U_t \Sigma_t^{-1} U_t^\mathsf{T} \preceq Y_t, \tag{C.7b}$$

$$\dot{\Sigma}_t = A_t \Sigma_t + \Sigma_t A_t^\mathsf{T} + B_t U_t + U_t^\mathsf{T} B_t^\mathsf{T} + D_t D_t^\mathsf{T}, \tag{C.7c}$$

$$\dot{\mu}_t = A_t \mu_t + B_t v_t, \tag{C.7d}$$

after noting that the constraint (C.7b) can be cast as a Linear Matrix Inequality (LMI) using Schur's complement as

$$\begin{bmatrix} \Sigma_t & U_t^\mathsf{T} \\ U_t & Y_t \end{bmatrix} \succeq 0.$$

Problem (C.7) is still infinite dimensional since the decision variables are functions of time $t \in [0,1]$; however, it can be discretized, approximately using a first-order approximation of the derivatives in (C.7c), (C.7d) (Chen et al., 2015b) or exactly using a zero-order hold (Liu et al., 2025; Rapakoulias & Tsiotras, 2023), and solved to global optimality using a semidefinite programming solver such as MOSEK (Mosek, 2020).

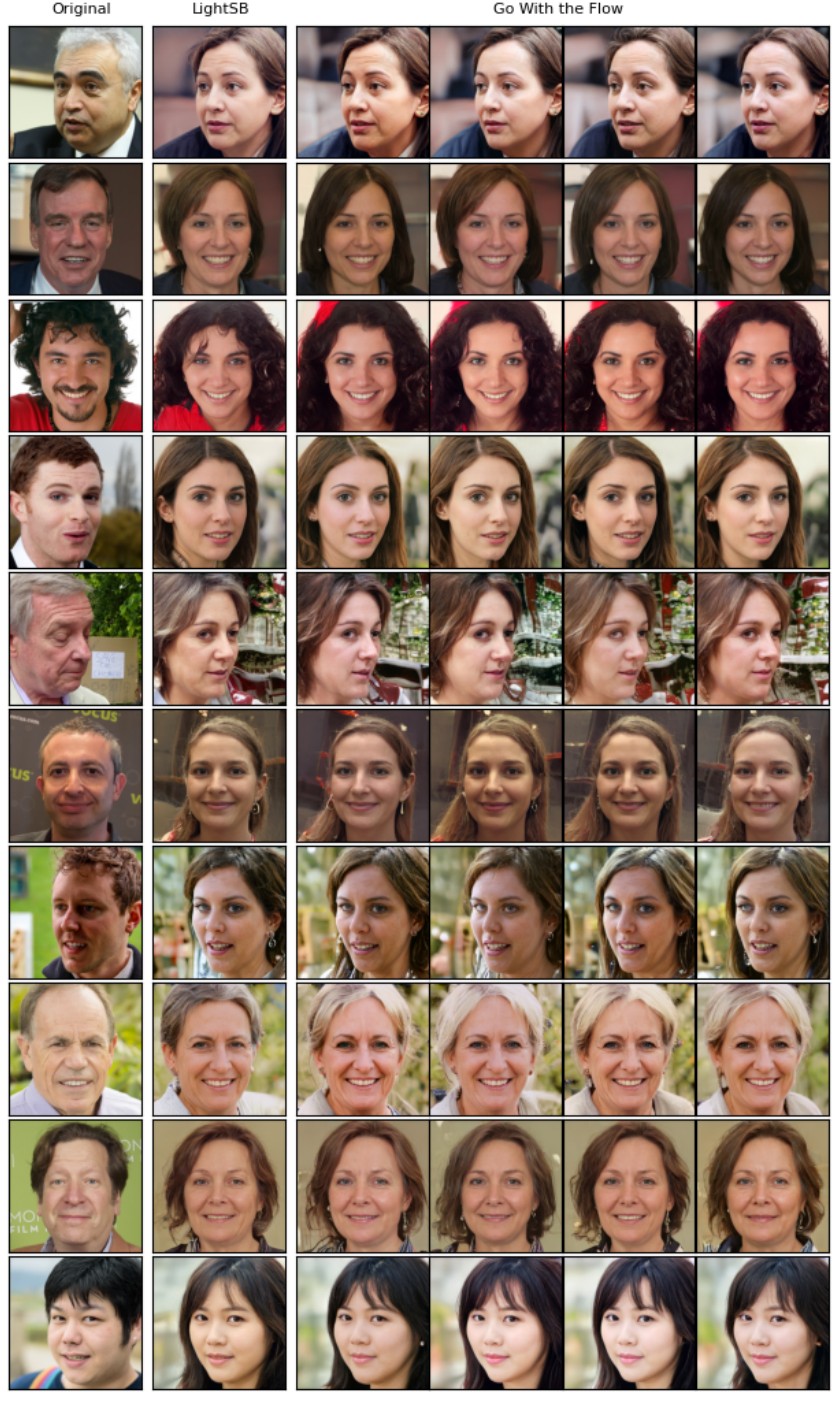

Figure 9: Further examples for the man-to-woman Image-to-Image translation task.

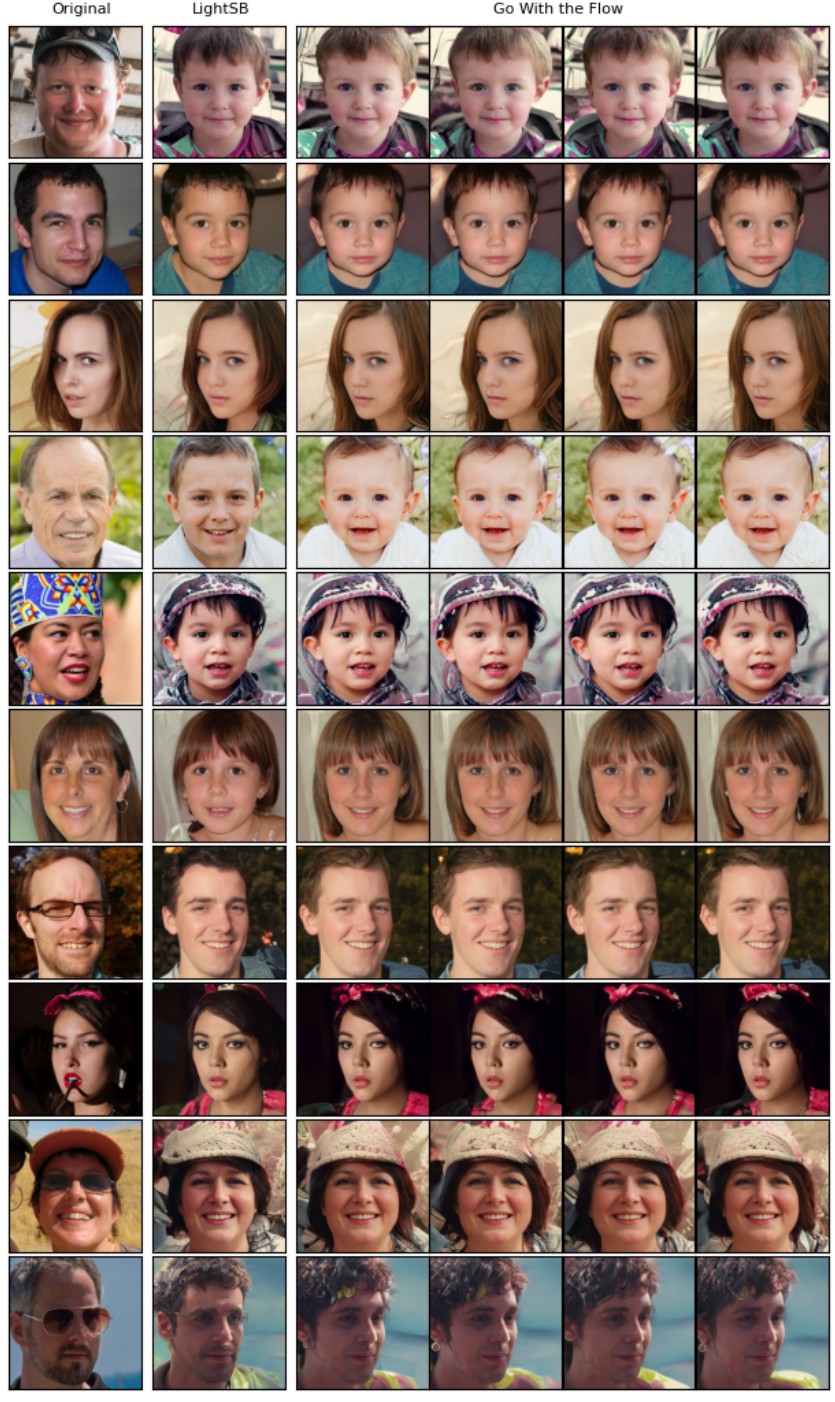

Figure 10: Further examples for the adult-to-child Image-to-Image translation task.

