# OpenReview forum: "Go With the Flow: Fast Diffusion for Gaussian Mixture Models"
_NeurIPS.cc/2025/Conference — NeurIPS 2025 spotlight_

### Official Review · Reviewer_VvBB · 2025-06-30

**Clarity:** 4
**Significance:** 2
**Originality:** 3
**Rating:** 5
**Confidence:** 4

**Summary:**

The authors proposes a training‑free method to solve Schrödinger Bridge problems when both endpoints are Gaussian Mixture Models (GMMs). The method expresses a global policy as a weighted mixture of closed‑form gaussian bridges, and chooses weights via a low‑dimensional linear program whose size grows with the number of mixture components. The paper gives convincing numerical results on a set of toy and real world problems including image-to-image translation.

**Questions:**

- The results for image-to-image translation tasks are compelling. Moreover it is surprising that you can get away with only a 10-component diagonal GMM. Do the authors imagine you do full image generation on some constrained datasets in a latent space? At what point does the method break down? It would be interesting to see the GM assumptions pushed to its limits.

*Possible action items (see weaknesses above)*
- Runtime analysis in terms of number of components, number samples, etc. (what ever is most relevant) for both training and inference.
- Algorithm block
- Error vs number of mixture components.

(I am not insisting that all of these must be done, I understand time constraints, but I do believe they would improve the paper.)

**Ethical Concerns:**

["NO or VERY MINOR ethics concerns only"]

**Final Justification:**

The work is strong and well presented. The approach is reasonable. A good edition to the conference.

**Limitations:**

- The author discuss, but obvious limit is the Gaussian assumption.

**Quality:**

3

**Strengths And Weaknesses:**

*Strengths*
- The paper is clear and easy to read.
- Training-free methods and Gaussian Mixture Models are probably under-rated in a world dominated by expensive generative methods. I like to see this kind of paper.
- The paper server as a good introduction to the area
- The derivation of the method is clear and the theory is substantial.
- The results for image-to-image translation tasks are compelling.

*Weaknesses*
- The clarity of the paper would be enhanced with an Algorithm block describing the full implementation.
- It would be valuable to know more about the sensitivity when boundary data are poorly approximated by a GMM. It would be great to see an experiment with error on the y-axis and number of GMM components on the x-axis for some of the examples. Is most of the error induced by fitting the marginals, or even if the marginal are well approximated by the GMM can other sources of error dominate?
- The paper is sold as a "fast" method yet there is no runtime analysis, empirical or theoretical. This type of analysis would shed light on key questions such as how does the runtime scale with dimension and number of samples etc. Similarly an experiment with runtime on the y-axis and components on the x-axis would be useful.

---

> ### Author Rebuttal · Authors · 2025-07-30
>
> We would like to sincerely thank the Reviewer for their thoughtful and constructive feedback. We are especially grateful for the recognition of the clarity of our presentation and the potential of training-free approaches.
> We will address all points raised by the Reviewer in detail:
>
>
> 1) **Regarding the use of 10-component diagonal GMMs in image-to-image translation tasks**: We appreciate the reviewer’s observation and insightful question. In our image-to-image translation experiments, both the input and output images are first mapped from their high-dimensional pixel space to a low-dimensional latent space of dimension 512 using a pretrained encoder. The GMM fitting and flow matching are then performed entirely in this latent space, where the distributions tend to be significantly smoother and more structured than in the original pixel domain. This dimensionality reduction, along with the regularity of the learned latent representations, helps explain why a relatively small number of diagonal components (e.g., 10) is sufficient to capture the essential modes of the data.
>
>
> 2) **Runtime analysis**  We thank the Reviewer for suggesting we add a theoretical analysis of the computational complexity of our algorithm to our work. The computational complexity of fitting a GMM using the Expectation-Maximization (EM) algorithm scales as $O (I N  K  (D + D^2))$ [1], where $I$ is the number of EM iterations, $N$ is the number of data points, $K$ is the number of Gaussian components (modes), and $D$ is the dimensionality of the data. This complexity highlights that GMM fitting becomes expensive with higher dimensions or a large number of modes. Once the GMMs are fitted, our method solves (LP) problem with $N_0 \times N_1$ variables, where  $N_0$ and $N_1$ denote the number of modes in the initial and final distribution, respectively. Modern solvers like MOSEK or Gurobi efficiently solve LP problems using interior-point methods, which have a computational complexity of $ O(\sqrt{l} N^3)$ [2], where  $l$ represents the number of constraints. This ensures scalability for moderate numbers of modes, but it can become computationally demanding as the number of modes in the GMMs exceeds $1,000$ for each boundary distribution. In practice, in the 2D example of our paper (see Figure 2 and Figure 6), we tested the algorithm in problems with 500-component GMMs and validated that training takes under a minute.
>
> Regarding the computational complexity of inference, each evaluation of the GMMflow policy (Equation 13) scales linearly with the number of components in each mixture and the SDE integration also scales linearly with the number of discretization time steps. In practice, when implementing the GMMflow policy, we only compute a small fraction of conditional GSB policies since the component-level transport plan is very sparse. Moreover, this computation is done in parallel for all conditional policies together. This results in very fast, practically constant-time inference regardless of the number of components in the boundary GMMs or problem dimension.
>
> To further benchmark our algorithm in practice, we provide wall clock times for both training and inference as a function of the number of components in the boundary distributions (same number for initial and final) and the problem dimensionality. We use the boundary distributions provided in the EOT benchmark for this test. We report these values here:
>
> **Training times in seconds**
>
>  | Dim\ \# comp | 5      | 10     | 20     | 50     | 100     |
>  |------------|--------|--------|--------|--------|---------|
>  | 2          | 0.209  | 0.0595 | 0.1083 | 0.2303 | 1.0256  |
>  | 16         | 0.0567 | 0.0948 | 0.1413 | 0.4218 | 3.3918  |
>  | 64         | 0.182  | 0.2228 | 0.7217 | 1.2875 | 2.1431  |
>  | 128        | 0.2802 | 0.4562 | 0.7713 | 1.6164 | 3.4101  |
>
>
> **Inference times in seconds**
>
>  | Dim\ \# comp | 5     | 10    | 20    | 50    | 100    |
>  |------------|-------|-------|-------|-------|--------|
>  | 2          | 0.037 | 0.031 | 0.031 | 0.032 | 0.030  |
>  | 16         | 0.032 | 0.032 | 0.032 | 0.033 | 0.032  |
>  | 64         | 0.032 | 0.032 | 0.032 | 0.032 | 0.039  |
>  | 128        | 0.032 | 0.033 | 0.036 | 0.032 | 0.082  |
>
> 3) **Algorithmic block** In response to the suggestion to improve clarity with an algorithm block, we have now added the following algorithm, describing the full implementation,  to the revised version of the paper.
>
>
> **Go With the Flow Algorithm**
>
> **Input** Initial dataset $\\rho_0$, final dataset $\rho_1$; number of GMM components $N_0$ and $N_1$.
>
> **Step 1:** $ \\{\alpha_0^{i}, \mu_0^i, \Sigma_0^i \\}_{i=1}^{N_0}\leftarrow$EM$(\rho_0, N_0)$ (Fits a GMM to initial dataset)
>
> **Step 2:** $ \\{\alpha_1^{j}, \mu_1^j, \Sigma_1^j \\}_{j=1}^{N_1}\leftarrow$ EM$(\rho_1, N_1)$ (Fits a GMM to terminal dataset)
>
> **Step 3:** $\forall (i,j)\in N_0 \times N_1$,
>     $\\{u_{t|ij}, \rho_{t|ij}, J_{ij}\\}  \leftarrow$ CS $((\mu_0^i, \Sigma_0^i),( \mu_1^j, \Sigma_1^j))$ (solves the $(i,j)$-th conditional GSB)
>
> **Step 4:** $\lambda_{ij} \leftarrow $  solve (15) using $ \\{J_{ij}, \alpha_0^{i}, \alpha_1^{j}\\}$
>
>
> **Step 5:**  Return $ \\{u_{t|ij}, \rho_{t|ij}, \lambda_{ij} \\}$
>
>
> 4) **Sensitivity of GMMs**: We agree that understanding the sensitivity of our method with respect to the quality of GMM approximation is an important direction. Although we have not conducted a dedicated sensitivity analysis experiment, our empirical results (e.g., in image translation tasks) suggest that relatively low-component GMMs can still produce compelling results, indicating a good degree of robustness. That said, we acknowledge that performance may degrade under severe underfitting, and a more systematic study of this sensitivity is a valuable avenue for future work.
>
>
> [1] Pedregosa, Fabian, et al. “Scikit-learn: Machine learning in Python.” Journal of Machine Learning
> Research 12 (2011): 2825-2830.

---

> > ### Comment · Reviewer_VvBB · 2025-08-01
> >
> > I thank the authors for their detailed feedback. I feel all my concerns have been addressed.
> >
> > The main draw back of the method appears to be the bad scaling with the number of dimensions. It would be interesting to see future work which attempts to address this problem directly with further dimensionality reduction, or some smart heuristic approximations to get around the quadratic scaling in D.
> >
> > None the less I like the paper and believe it is a reasonable addition to the conference.

---

> > > ### Author Response · Authors · 2025-08-03
> > > **Thanks for the in-depth review.**
> > >
> > > We sincerely thank the Reviewer for their time in providing a high-quality, in-depth, and positive evaluation of our work. We believe their comments helped in improving the quality of our submission.

---

### Official Review · Reviewer_kjHY · 2025-07-01

**Clarity:** 2
**Significance:** 2
**Originality:** 3
**Rating:** 4
**Confidence:** 3

**Summary:**

The authors address the Schrödinger Bridge (SB) problem for Gaussian mixture endpoints (SB-GM). They observe that the SB between two individual Gaussians (SB-G) admits a closed-form solution, and hence propose to parametrize the feasible SB-GM solutions as a weighted mixture of SB-G solutions. By doing so, they mitigate the original optimization problem and optimize only over the mixture weights to receive the feasible solution. They the authors define a linear optimization problem for optimization over the mixture weights and rigorously show that the linear optimization problem objective upper-bounds the true optimal cost for the SB-GM. Building on this insight, the authors extend their framework to both the multi-marginal momentum SB and to SBs between infinite Gaussian mixtures.

In practice, they approximate arbitrary distribution endpoints $p_0$ and $p_1$ by finite Gaussian mixtures, construct SB-G solutions for every pair of mixture components, and recover the approximation of overall SB-GM by solving a linear optimization problem over mixture weights.  An analogous scheme is developed for the multi-marginal momentum SB. The method is validated on 2D toy problem, Image-to-Image translation in the latent space of ALAE and learning dynamics of cell populations.

**Questions:**

- Is the optimal solution for the SB problem with gaussian mixture endpoints lies in the set of solutions defined in Theorems 1, 3, 5?
- Is the $J_{\text{OT}}$ in Theorems 2, 4, 6 transport cost for the corresponding $\lambda$ defined solutions?
- Are there any results on tightness of upper bounds in Theorems 2,4,6?
- What is the computational complexity of the linear optimization algorithms used, or that could be potentially used? Including worst cases. Is there any results on training time and method quality w.r.t. number of gaussian mixtures used for approximation (preferably on EOT bench)?
- Is there results on transport cost for the ALAE experiment?
- Why do you have relatively high SWD and very low MMD on cell population dynamics learning experiment?
- How do you apply NLSB [5] to multi-marginal SB problem in the cell population dynamics learning experiment? As far as I understand NLSB is general SB method. If you consider the general SB method, then I think you also should consider methods [1, 2, 3, 4].
- What potential algorithms can be considered for solving the stated in Theorem 6 optimization problem?

**Ethical Concerns:**

["NO or VERY MINOR ethics concerns only"]

**Final Justification:**

The authors have adressed my concerns.

**Limitations:**

Mostly yes, but I would like authors to clarify the complexity of linear optimization methods they use w.r.t. number of components in Gaussian Mixtures. Depending on the answer, there is a possibility one could consider it a limitation.

**Paper Formatting Concerns:**

No major formatting issues in this paper.

**Quality:**

2

**Strengths And Weaknesses:**

**Strengths**:

- The proposed algorithm is both novel and computationally efficient, demonstrating strong performance in low-dimensional settings.
- The authors’ approximation strategies for diverse SB formulations—including the multi-marginal momentum SB and the infinite-mixture SB—represent valuable conceptual contributions.
- Empirical results on image-to-image translation, EOT benchmark (Appendix C)  and cell-population dynamics modeling show that GMMflow mostly matches state-of-the-art methods.
- The proposed algorithm tries to advance the area of SB methods specifically crafted for low dimensional cases.

**Weaknesses**:

- Authors do not provide guarantees for a mixture of SB-G solutions to reach the optimal solution for the SB-GM problem. In general, only the upper bound on the optimal transport cost can be received that way. For the general SB the approximation of its distribution endpoints by the gaussian mixtures is already an approximation and the mixture of SB-G solutions adds a second approximation on top.
    - Would be nice to know more about the solutions gotten as the result of solving linear optimization problems in Theorems 2, 4, 6 and their relation to the optimal ones. Maybe there is some cases where you can find exactly the optimal solutions. For other related questions see **Questions** section.
- The number of variables in the linear optimization problem (mixture weights) grows linearly with the number of components in Gaussian Mixtures used for approximation of distributions. In the two distribution case number of variables is: $(M_1 \times M_2)$, or in the multi-marginal case  $(M_1 \times M_2 … \times M_k)$. This seems like a lot of mixture weights, especially when one considers complex distributions that require a lot of Gaussian Mixture components to be approximated accurately. Additionally one has to solve the linear optimization problem, with its own computational complexity, to optimize over these mixture weights.
    - Would like to see a study on the growth of Gaussian Mixtures components along with  training time and method quality.
- As far as I understand the inference of the SB-G mixture solution requires the simulation of the SDE. Which is rather unconvenient, especially since LSB/LSBM and Gaussian to Gaussian SB solution transport plans can be inferred without the SDE simulation.
- The evaluation:
    - In the Image-to-Image translation experiment only the target distribution matching  metrics (FID, ALAE-BW) are shown. Which is weird since the target matching is guaranteed, up to GMM approximation error, by the construction of the method. On the other hand, the optimality of transport cost is not guaranteed. Notably by the Figure 3 one can see that LightSB transport plan has lower transport cost.
    - The only experiment that quantitatively evaluates the transport cost is EOT benchmark shown in Appendix C. The performance of method becomes better with growth of $\epsilon$, while the lower $\epsilon$ examples do mostly evaluate the transport cost of the method and higher $\epsilon$ examples do mostly evaluate the ability of method to fit the ending ($p_1$) distribution. This experiment results can be seen as proof that the proposed method is bad as transport cost but good at distribution fitting.
    - In the EOT benchmark experiment some baselines are missing: DSBM [1], SF2M-sink [2], LightSB [3], LightSBM [4]. All the metrics are already computed, see Table 1 [4].

**Typos/Minor text concerns**:

- Line 128: $p_i, p_f$ should it be $p_0, p_1$?
- Line 298, 300: Bures-Wasserstein distance
- Line 332, 337, Table 3: NSBL → NLSB

[1] Shi, Y., De Bortoli, V., Campbell, A., & Doucet, A. (2023). Diffusion schrödinger bridge matching. *Advances in Neural Information Processing Systems*, *36*, 62183-62223.

[2] Tong, A. Y., Malkin, N., Fatras, K., Atanackovic, L., Zhang, Y., Huguet, G., ... & Bengio, Y. (2024, April). Simulation-Free Schrödinger Bridges via Score and Flow Matching. In *International Conference on Artificial Intelligence and Statistics*(pp. 1279-1287). PMLR.

[3] Korotin, A., Gushchin, N., & Burnaev, E. Light Schrödinger Bridge. In *The Twelfth International Conference on Learning Representations*.

[4] Gushchin, N., Kholkin, S., Burnaev, E., & Korotin, A. (2024, July). Light and optimal schrödinger bridge matching. In *Forty-first International Conference on Machine Learning*.

[5] Koshizuka, T., & Sato, I. Neural Lagrangian Schr\"{o} dinger Bridge: Diffusion Modeling for Population Dynamics. In *The Eleventh International Conference on Learning Representations*.

---

> ### Author Rebuttal · Authors · 2025-07-30
>
> We thank the reviewer for the careful consideration of our work, for finding our results novel and valuable conceptual contributions, and for recognising the computational efficiency of our method. We will address all the comments raised by the reviewer in detail. If all the Reviewer's concerns are well addressed, we would appreciate a positive reflection in his/her's score.
>
> 1. **Optimality of our approach and tightness of the upper bound in Theorem 2**: As mentioned in Theorems 1, 3, and 5, the mixture policy in equation (13) introduces a family of **feasible** policies for the SB, which is smaller than the class of all feasible solutions for the SB. In general, the globally optimal solution of the SB might not belong to this family, which introduces some suboptimality with respect to the transport cost, in exchange for numerical tractability. To empirically quantify this optimality gap in our original submission, we provide extensive benchmark testing in Appendix C.4 in Table 6. In practice, we observe that although the mixture policy calculated by Theorems 1, 2 may, in general, be suboptimal, in practice results in better coupling and transport costs, especially for larger values of the noise parameter $\epsilon$, than most neural SB or EOT methods, due to the non-convex loss functions associated with the corresponding methods. To improve the clarity of our work and strengthen our contribution, **we have added a new result to our paper**, which explores the suboptimality introduced by the upper bound of Theorem 2. This new result shows that when the components of the GMMs are well-separated, the upper bound introduced in Theorem 2 is tight, and the mixture policy is optimal within the feasible set introduced in Theorem 1. Due to space limitations, and to avoid a duplicate response, we refer to point 2. in our response to Reviewer HrsH for the rigorous proof of this statement.
>
> 2. **$J_{OT}$ and transport cost of $\lambda$ policy** As Theorem 2 suggests, the cost $J_{OT}$ is an upper bound of the true transport cost of the mixture policy (13). This bound becomes tight when the components of the solution are well-separated. In general, using the mixture policy is guaranteed to yield a cost that is at least smaller than the upper bound $J_{OT}$. This can also be validated experimentally, for example, through Table 4 (see columns 1st and 2nd columns for the upper bound $J_{OT}$ and the true cost $J_{GMM}$ respectively).
>
> 3. **Computational complexity of solving Linear Programs (LPs)** We thank the Reviewer for this comment. Modern solvers like MOSEK or Gurobi efficiently solve LP problems using interior-point methods, which have a computational complexity of $ O(\sqrt{l} N^3)$ [6], where  $N$ is the number of decision variables and $l$ represents the number of constraints. In practice, this allows solving LPs with millions of variables easily. In practice, in the 2D example of our paper (see Figure 2 and Figure 6), we tested the algorithm in problems with 500-component GMMs (250,000 decision variables in the corresponding LP),  and validated that training takes under a minute for both GMM fitting and solving the LP. For a full complexity analysis of our method, we refer the Reviewer item 4. in our response to Reviewer PeSU.
>
> To further benchmark our algorithm in practice, we provide wall clock times for both training and inference with respect to the number of components and the problem dimensionality for the boundary distributions provided in the EOT benchmark. We report these values here:
>
>  **Training times in seconds**
>
>  | Dim\ \# comp | 5      | 10     | 20     | 50     | 100     |
>  |------------|--------|--------|--------|--------|---------|
>  | 2          | 0.209  | 0.0595 | 0.1083 | 0.2303 | 1.0256  |
>  | 16         | 0.0567 | 0.0948 | 0.1413 | 0.4218 | 3.3918  |
>  | 64         | 0.182  | 0.2228 | 0.7217 | 1.2875 | 2.1431  |
>  | 128        | 0.2802 | 0.4562 | 0.7713 | 1.6164 | 3.4101  |
>
>  **Inference times in seconds**
>
>  | Dim\ \# comp | 5     | 10    | 20    | 50    | 100    |
>  |------------|-------|-------|-------|-------|--------|
>  | 2          | 0.037 | 0.031 | 0.031 | 0.032 | 0.030  |
>  | 16         | 0.032 | 0.032 | 0.032 | 0.033 | 0.032  |
>  | 64         | 0.032 | 0.032 | 0.032 | 0.032 | 0.039  |
>  | 128        | 0.032 | 0.033 | 0.036 | 0.032 | 0.082  |
>
> We note that when implementing the GMMflow policy in practice, we only compute a small fraction of conditional GSB policies since the component-level transport plan is very sparse. Moreover, this computation is done in parallel for all conditional policies together. This results in very fast, practically constant-time inference regardless of the component number or problem dimension.
>
> Finally, we note that because the EOT benchmark uses an initial Gaussian distribution and a GMM terminal distribution, there is no point in reporting metrics such as marginal distribution accuracy or transport plan optimality as a function of the number of GMM components, since these will perform best when the number of components used in GMMflow matches the setting of EOT benchmark. Furthermore, exploring how well a GMM approximates a general distribution as the number of components increases is a well-studied problem and goes beyond the scope of our work, so due to time constraints, we will not provide experiments that explore this issue.
>
> 4. **Evaluation of the transport cost for the ALAE experiment** To further evaluate the transport cost of our method, we will calculate the empirical Wasserstein transport cost of our method and compare it with LSB and LSBM. More specifically, we will report the values of $\big( \int\|x_0-x_1\|^2 \mathrm{d} \pi(x_0, x_1) \big)^{\frac{1}{2}}$ for different levels of the regularization constant $\epsilon$, using the transport plan from GMMflow, LSB and LSBM. We used 10,000 samples to calculate the empirical expectation.
>
>  |M2W             | GMMflow | LSB  | LSBM   |
>  |----------------|---------|------|--------|
>  |$\epsilon=0.1$  | 11.14   |10.47 |	10.44|
>  |$\epsilon=0.01$ | 9.05    |8.23  | 8.18   |
>  |$\epsilon=0$    | 8.75    |-     | -      |
>
>  |A2C             | GMMflow | LSB  | LSBM   |
>  |----------------|---------|------|--------|
>  |$\\epsilon=0.1$  |11.41    |10.45	| 10.43  |
>  |$\epsilon=0.01$ |9.33	 |8.19	| 8.18   |
>  |$\epsilon=0$    | 9.04    |  -   |  -     |
>
> We note that although this experiment agrees with the intuition of the Reviewer, namely that GMMflow results in higher transport costs than LSB/LSBM, since the difference is small, it is hard to attribute this to any potential suboptimality introduced by the upper bound of Theorem 2 or the restriction of the SB policy in the feasible set introduce by Theorem 1. Specifically, the 40\% lower FID and 70\% lower ALAE-BW metrics in Table 1 of the paper indicate that GMMflow learns considerably different marginal distributions than LSB/LSBM. Therefore, it is not clear whether the increased transport costs are due to the different marginal distributions or the suboptimal transport plan.
>
> 5. **High SWD and very low MMD**: We thank the reviewer for this observation. We attribute this to the nature of using GMMs to capture complicated distributions. The very low MMD indicates the accurate fitting of GMMs, since the MMD metric is moment-based and captures the low-order information in the distributions. On the other hand, since the true distributions are not GMMs, the SWD metric shows that some higher-order information is lost.
>
> 6. **NLSB algorithm in the RNA example**. The Neural Lagrangian Schr\"odinger Bridge (NLSB) solver is more complicated than a standard SB solver since the cost function it optimizes contains density-dependent and velocity regularization terms. We refer the Reviewer to [5] for the NLSB implementation details in the cellular dynamics example. We argue that existing implementations of methods [1-4] are not directly applicable to the multimarginal setting unless the problem is solved pairwise for any two consecutive distributions, reducing the problem to a series of standard, two-marginal SBs.
>
> 7. **Infinte component GMM computation**. While solving the optimization problem (24) in Theorem 6 is challenging, in general, the scope of our work focuses on the special case where the parameter spaces $w_0, w_1$ are 1-dimensional. As detailed in Appendix B of our supplementary material, this contains multiple important classes of problems, such as the computation of the Wasserstein distance between multivariate Student-t distributions. In this setting, Problem (24) can be solved in closed form using Proposition 3 from the supplementary material, yielding a comprehensive upper bound in the Wasserstein distance between Student-t distributions.
>
> 8. **Minor Typos** We thank the reviewer for pointing these out. We will correct these typos in the final version of our work.
>
>
> [1] Shi, Y., De Bortoli, V., Campbell, A., \& Doucet, A. (2023). Diffusion schrödinger bridge matching. Advances in Neural Information Processing Systems, 36, 62183-62223.
>
> [2] Tong, A. Y., Malkin, N., Fatras, K., Atanackovic, L., Zhang, Y., Huguet, G., ... \& Bengio, Y. (2024, April). Simulation-Free Schrödinger Bridges via Score and Flow Matching. In International Conference on Artificial Intelligence and Statistics(pp. 1279-1287). PMLR.
>
> [3] Korotin, A., Gushchin, N., \& Burnaev, E. Light Schrödinger Bridge. In The Twelfth International Conference on Learning Representations.
>
> [4] Gushchin, N., Kholkin, S., Burnaev, E., \& Korotin, A. (2024, July). Light and optimal schrödinger bridge matching. In Forty-first International Conference on Machine Learning.
>
> [5] Koshizuka, T., \& Sato, I. Neural Lagrangian Schr"{o} dinger Bridge: Diffusion Modeling for Population Dynamics. In The Eleventh International Conference on Learning Representations.
>
> [6] Pedregosa, Fabian, et al. “Scikit-learn: Machine learning in Python.” Journal of Machine Learning
> Research 12 (2011): 2825-2830

---

> > ### Comment · Reviewer_kjHY · 2025-08-03
> >
> > I thank the authors for taking the time to answer my questions and address my concerns. Especially, I am pleased to see the result on the tightness of the upper bound from Theorem 2. Please incorporate this and other results regarding experimental evaluation (Answers 3 and 4) into the final version of the paper. In that light, I raise my score up to 4.

---

> > > ### Author Response · Authors · 2025-08-03
> > > **Thanks for the in depth review**
> > >
> > > We would like to thank the Reviewer for their in-depth evaluation of our work and the careful consideration of our response!

---

### Official Review · Reviewer_nHAK · 2025-07-03

**Clarity:** 4
**Significance:** 4
**Originality:** 3
**Rating:** 5
**Confidence:** 3

**Summary:**

The authors provide a specific formulation of optimal transport that is useful in low-dimensional regimes and avoids the computationally complexity suffered by more established models in high-dimensional regimes. In particular, they propose a policy which is theoretically feasible and an associated probability flow for the setting of transportation between mixed Gaussian conditionals, by solving Gaussian bridge sub-problems. They support their theoretical findings with experiments comparing their methodology to others' on toy data, image matching, and single-cell data with their methodology outperforming contemporary methods in every instance.

**Questions:**

N/A

**Ethical Concerns:**

["NO or VERY MINOR ethics concerns only"]

**Final Justification:**

The authors have sufficiently defended their work and the paper is deserving of acceptance.

**Limitations:**

Yes, the authors have addressed limitations. The other limitations are included in the weaknesses mentioned in the previous paragraph.

**Quality:**

4

**Strengths And Weaknesses:**

The authors concisely summarize contemporary methods for optimal transport between probability distributions and describe the gap their paper intends to close. They provide theoretical evidence supporting their proposed transport plan and make a clear delineation between existing methods and what their method offers computationally.  The empirical results are strong; their proposed transport plan outperforms contemporary methods across a diverse set of experiments across different metrics. A potential improvement, as mentioned by the authors, is to not constrain the proposed methodology to the setting of Gaussian Mixture Models. It could also be worth empirically exploring around what dimension the proposed model loses its efficacy through ablation studies.

---

> ### Author Rebuttal · Authors · 2025-07-30
>
> We sincerely thank the Reviewer for the positive evaluation of our work and for the suggestion to further test our algorithm through ablation studies to study how it scales in practice on problems with many components. To address this suggestion, we performed a run-time analysis for training and inference with respect to the problem dimension and nubmer of GMM components. To avoid duplicating the response, we refer the Reviewer to point 4. in our response to Reviewer PeSU.

---

### Official Review · Reviewer_PeSU · 2025-07-03

**Clarity:** 3
**Significance:** 2
**Originality:** 2
**Rating:** 5
**Confidence:** 4

**Summary:**

The paper develops a computational method to compute suboptimal solutions of the schrödinger bridge problem between gaussian mixture models, and showcases the potential of the problem in low-dimensional settings. The key idea is to exploit the decoupling of particles dynamics with the assignment problem, building on top of recent work on the topic.

**Questions:**

1. Can the authors discuss further their suboptimality gap? Related: How does the approximation of GMM further exacerbate this gap?
2. Can the authors provide comparisons (at least in the deterministic case) with the approach in https://epubs.siam.org/doi/abs/10.1137/23M1560902?
3. Can the authors clarify the setting for the RNA dataset?

**Ethical Concerns:**

["NO or VERY MINOR ethics concerns only"]

**Final Justification:**

The authors addressed all my concerns and accordingly I think the paper deserves acceptance.

**Limitations:**

The authors adequately discuss the limitations.

**Paper Formatting Concerns:**

I do not see any issue.

**Quality:**

3

**Strengths And Weaknesses:**

The paper is very focused on a specific problem with clearly stated limitations. The theory is correct (beside a few minor clarifications needed) and gradually introduces more complexity, rendering the paper very pleasant to read. I identified the following weaknesses, that the authors may consider to address to strengthen their contribution:

1. Although reading the entire paper makes the scope of the work very clear, I believe it could be stated better in the introduction: low-dimensional problem in which the marginals can be well approximated by GMM, and in which there is a need for a good solution, fast, but not necessarily optimal. To build on this, the authors could also exemplify a few of these applications.
2. In the contributions, it is stated that the paper proposes a solution to the SB problem with GMM, but technically, it only provides a suboptimal solution. Moreover, the suboptimality gap is not quantified and may be very large. Carefully addressing this gap or clarifying this potential limitation will strengthen the paper for other people to build on top.
3. The authors cite https://epubs.siam.org/doi/abs/10.1137/23M1560902, but perhaps they could also contextualize their result with respect to the recipe provided in https://epubs.siam.org/doi/abs/10.1137/23M1560902. Namely, the mixing discussed in Go with the Flow is an approximation of the mixing derived in https://epubs.siam.org/doi/abs/10.1137/23M1560902, hence the suboptimality. Note that the work cited is for deterministic case, another suboptimality comes from the stochastic dynamics in this type of mixing. When comparing with the literature, in the deterministic case, they could consider comparing to https://epubs.siam.org/doi/abs/10.1137/23M1560902, which provides an exact solution. They could compare it also in the stochastic case, to have an idea of a different cost approximation.
4. The suboptimality of the derived bound is unclear. Can the authors comment on it?
5. The efficiency benefits are somewhat minor (the algorithms considered are all very fast), so it is hard to tell if the benefits come from algorithmic speedups or from implementation details without a complexity analysis.
6. The setting for the RNA dataset is not very clear (it is almost never clear in the literature either….). Can you please clarify the objective, how do you evaluate, etc? Also, for the experiments, while I appreciate the numerous experiments I believe that it would be better to defer more to the appendix and discuss more thoroughly less experiments in the main text.

Some other comments:

1. The authors present 6 theorems. While I appreciate the increasing complexity, reversing the order of the theorem would allow to discuss some of the results as corollary and gain some space for more thorough discussions in the main text. It would also allow to avoid essentially duplicated proofs. I would also make a Lemma instead of a Theorem for the first theorem of each “pair” of theorems.
2. Can you please double check Proposition 1? The expression for the velocity field seems wrong (independent of t). Or could the authors perhaps clarify?
3. Before equation 11, the authors refer to Problem 6 and 10, but then mention replacing the first constraint only in 6 (typo? perhaps you want to add 10 as well). I would also not use the word sequel as it made me think of a follow up paper.
4. Perhaps stop the sentence “Problem (24) is challenging to solve in general.” instead of the “,”.
5. By the way, I find this decoupling approach very interesting. It also relates to a recent work on a similar, yet slightly different setting https://arxiv.org/abs/2406.12616. There the decoupling is somewhat dual (for estimation).

---

> ### Author Rebuttal · Authors · 2025-07-30
>
> We would like to sincerely thank the Reviewer for his/hers positive and in-depth evaluation of our work, for finding our work pleasant to read, and for the multiple sincere suggestions to improve the quality and presentation of our work. We will address all issues raised by the reviewer in detail.
>
>    1. **Suboptimality gap of our method**. As mentioned in Theorems 1, 3, and 5, the mixture policy in equation (13) introduces a family of feasible policies for the SB, which is smaller than the class of all feasible solutions for the SB. In general, the globally optimal solution of the SB might not belong to this family, which introduces some suboptimality with respect to the transport cost, in exchange for numerical tractability. To empirically quantify this optimality gap in our original submission, we provide extensive benchmark testing in Appendix C.4 in Table 6. In practice, we observe that although the mixture policy calculated by Theorems 1, 2 may, in general, be suboptimal, in practice results in better coupling and transport costs, especially for larger values of the noise parameter $\epsilon$, than most neural SB or EOT methods, due to the non-convex loss functions associated with the corresponding methods. To improve the clarity of our work and strengthen our contribution, **we have added a new result to our paper**, which explores the suboptimality introduced by the upper bound of Theorem 2. This new result shows that when the components of the GMMs are well-separated, the upper bound introduced in Theorem 2 is tight, and the mixture policy is optimal within the feasible set introduced in Theorem 1. To save space and avoid a duplicate response, we refer to point 2) in our response to Reviewer HrsH for the rigorous proof of this statement.
>
>    2. **Comparison with the approach in [1]** We thank the Reviewer for this suggestion. As the reviewer mentions, the cited work is for deterministic, discrete-time problems while our scope is continuous-time deterministic/stochastic problems. Furthermore, in order to apply the result of [1], one needs to be able to explicitly calculate the Wasserstein-2 optimal transport plan between Gaussian Mixture Models in order to implement the policy in [1, Example 5.2], which is not available in closed form for GMMs.
>    Therefore, the only alternative would be to use the empirical W-2 distance using samples. We note that for the 2D examples shown in Figure 1, we have already performed this calculation in Appendix C.1, Table 4 (see first line for the deterministic case).
>
>
>   3. **RNA experiment setup** We thank the reviewer for this comment. We will present a brief overview of the experiment here and add more details in the main text of the paper. From an optimization perspective, in the RNA example, we solve an instance of Problem (16). The dataset, i.e., the marginal distributions, consists of 5-dimensional distributions at 5 time steps, $t_1, ..., t_5$. In the distribution of each time step, we first fit a 5-component GMM to obtain the GMM parameters of the constraint (16c). To compute $\lambda_{\mathbf{i}}$ in Theorem 4, one first needs to calculate $J_{\mathbf{i}}$, which is the cost of each Gaussian Multi-marginal momentum Schrodinger Bridge (GMSB) of the form (10). While there is no closed solution for the GMSB, it can be efficiently calculated as a Semidefinite Program (see Appendix D for the details). Once the cost tensor $J_{\mathbf{i}}$ is computed, $\lambda_{\mathbf{i}}$ is computed by solving the corresponding Problem (20), which then yields the flow through Equation (18). To evaluate our results, besides visual means through Figures 4 and 7 (see supplementary material), we use the Sliced Wasserstein Distance (SWD) and the Maximum Mean Discrepancy (MMD) between the true marginal distributions and the inferred marginal distributions. To calculate the inferred distributions, we use the first marginal at time $t_1$ as an initial condition for the position variable, i.e., $x_1$, and infer the velocity by sampling from Equation (B.19d) (this is easy since (B.19d) is a GMM). We then integrate the flow in order to sample from the marginals at the rest of the time steps $t_2,...,t_5$. The MMD and SWD both measure distributional discrepancy between the inferred and target distributions in a given time step. We average the calculated SWD and MMD values for the four predicted time steps (we skip the first since it was used for initial conditions) and report the values at Table 3, along with other baselines.
>
> **Other comments**:
>
>    4. **Computational complexity of our method**. We thank the Reviewer for suggesting adding a theoretical analysis of the computational complexity to our work. The computational complexity of fitting a GMM using the Expectation-Maximization (EM) algorithm scales as $O (I N  K  (D + D^2))$ [1], where $I$ is the number of EM iterations, $N$ is the number of data points, $K$ is the number of Gaussian components (modes), and $D$ is the dimensionality of the data. This complexity highlights that GMM fitting becomes expensive with higher dimensions or a large number of modes. Once the GMMs are fitted, our method solves (LP) problem with $N_0 \times N_1$ variables, where  $N_0$ and $N_1$ denote the number of modes in the initial and final distribution, respectively. Modern solvers like MOSEK or Gurobi efficiently solve LP problems using interior-point methods, which have a computational complexity of $ O(\sqrt{l} N^3)$ [2], where  $l$ represents the number of constraints.   This ensures scalability for moderate numbers of modes, but it can become computationally demanding as the number of modes in the GMMs exceeds $1,000$ for each boundary distribution. In practice, in the 2D example of our paper (see Figure 2 and Figure 6), we tested the algorithm in problems with 500-component GMMs and validated that training takes under a minute.
>
> Regarding the computational complexity of inference, each evaluation of the GMMflow policy (Equation 13) scales linearly with the number of components in each mixture and the SDE integration also scales linearly with the number of discretization time steps. In practice, when implementing the GMMflow policy, we only compute a small fraction of conditional GSB policies since the component-level transport plan is very sparse. Moreover, this computation is done in parallel for all conditional policies together. This results in very fast, practically constant-time inference regardless of the number of components in the boundary GMMs or problem dimension.
>
> To further benchmark our algorithm in practice, we provide wall clock times for both training and inference as a function of the number of components in the boundary distributions (same number for initial and final) and the problem dimensionality. We use the boundary distributions provided in the EOT benchmark for this test. We report these values here:
>
> **Training times in seconds**
>
>  | Dim\ \# comp | 5      | 10     | 20     | 50     | 100     |
>  |------------|--------|--------|--------|--------|---------|
>  | 2          | 0.209  | 0.0595 | 0.1083 | 0.2303 | 1.0256  |
>  | 16         | 0.0567 | 0.0948 | 0.1413 | 0.4218 | 3.3918  |
>  | 64         | 0.182  | 0.2228 | 0.7217 | 1.2875 | 2.1431  |
>  | 128        | 0.2802 | 0.4562 | 0.7713 | 1.6164 | 3.4101  |
>
>
> **Inference times in seconds**
>
>  | Dim\ \# comp | 5     | 10    | 20    | 50    | 100    |
>  |------------|-------|-------|-------|-------|--------|
>  | 2          | 0.037 | 0.031 | 0.031 | 0.032 | 0.030  |
>  | 16         | 0.032 | 0.032 | 0.032 | 0.033 | 0.032  |
>  | 64         | 0.032 | 0.032 | 0.032 | 0.032 | 0.039  |
>  | 128        | 0.032 | 0.033 | 0.036 | 0.032 | 0.082  |
>
>   5. **Merging theorem proofs** We thank the reviewer for this suggestion. Although we believe that keeping the theorems separate makes the flow of the paper easier to follow, we agree that the proofs of Theorems 1,5, and Theorems 2, 6 are similar. We will consider merging them in the final version of our work for conciseness. The multi-marginal versions of the same theorems are considerably different, which prompts us to keep the proofs separate.
>
>   6. **Proposition 1 is correct** We thank the reviewer for this comment. We confirm that Proposition 1 is correct. The vector field $u_t(x) = K_t(x - \mu_t) + v_t$ is correct and will, in general, be time varying since $K_t$ and $\mu_t$ are time-varying. We note that for the Gaussian Schr\"odinger Bridge, the feed-forward term $v_t= \mu_1 - \mu_0$ is not time varying; however, we are keeping the dependence in time since for more general bridges, such as the Gaussian Momentum SB or the SB with general linear prior dynamics, this term will be time varying.
>
>   7. **Readability and scope** To increase readability and better convey the scope of our work, we have altered the abstract to better highlight our motivation, namely, the efficient solution of low-dimensional SB problems.
>   In the revised version of the manuscript, we have changed the sentence before Equation (6) to read "the first constraint in (6) or (10b).'' Furthermore, we will refrain from using "sequel'' to avoid confusion.
>
>
> [1] Pedregosa, Fabian, et al. ``Scikit-learn: Machine learning in Python." Journal of Machine Learning Research 12 (2011): 2825-2830.
>
> [2] Boyd, Stephen P., and Lieven Vandenberghe. Convex optimization. Cambridge University Press, 2004.

---

> ### Comment · Reviewer_PeSU · 2025-08-01
> **Thanks for the rebuttal.**
>
> Thanks for addressing my concerns.
>
> I have one more question:
>
> Could you clarify further the impact of the problem you want to solve? What are possible applications, why this is important for the community, etc? You promise to make it more clear in introduction and abstract, but it would be beneficial for me as well to justify a score increase.
>
> I would appreciate if the authors could motivate better this remaining point.

---

> > ### Author Response · Authors · 2025-08-03
> > **Authors response to additional question.**
> >
> > We thank the Reviewer for this thoughtful question.
> >
> > Our main motivation is to complement the fast-growing body of literature on neural approaches for solving Schrodinger Bridge (SB) and Optimal Transport (OT) problems, which have recently attracted significant attention due to their applications in many practical tasks in biology [1] and computer science [2]. As the Reviewer points out, our technical motivation is the proposal of a computationally inexpensive approach to approximately solve SB and OT problems with Gaussian Mixture marginals. In the same way GMMs play a key role in machine learning and statistics, with abundant applications in Bayesian inference and inverse problems [3], signal processing [4], and stochastic control [5], to name a few, we believe that our method will complement their use cases, as a handy tool for solving related transport problems, without requiring the extensive hyperparameter turning and computational resources for training more general neural-based algorithms.
> > More specifically, rapid SB solvers could find application in problems where a real-time solution is sought, such as in online decision making and multi-agent control paradigms. Such an approach would allow deploying closed-loop in a distribution level feedback policies, an active topic of research in mean-field control theory and swarm control [4].
> > Further applications could come from real-time monitoring and prediction of biological processes, where the objective is to infer the evolution of individuals given collective measurements.
> >
> > [1] Bunne, Charlotte, et al. "Optimal transport for single-cell and spatial omics." Nature Reviews Methods Primers 4.1 (2024): 58.
> >
> > [2] Peyré, Gabriel, and Marco Cuturi. "Computational optimal transport: With applications to data science." Foundations and Trends® in Machine Learning 11.5-6 (2019): 355-607.
> >
> > [3] Bouguila, Nizar, and Wentao Fan, eds. Mixture models and applications. Vol. 530. Berlin/Heidelberg, Germany: Springer, 2020.
> >
> > [4] Plataniotis, Kostantinos N., and Dimitris Hatzinakos. "Gaussian mixtures and their applications to signal processing." Advanced signal processing handbook (2017): 89-124.
> >
> > [5] Balci, Isin M., and Efstathios Bakolas. "Density steering of Gaussian mixture models for discrete-time linear systems." 2024 American Control Conference (ACC). IEEE, 2024.
> >
> > [6] Emerick, Max, Jared Jonas, and Bassam Bamieh. "Causal Tracking of Distributions in Wasserstein Space: A Model Predictive Control Scheme." 2024 IEEE 63rd Conference on Decision and Control (CDC). IEEE, 2024.

---

> > > ### Comment · Reviewer_PeSU · 2025-08-03
> > > **Thank you, I will raise my score accordingly.**
> > >
> > > Thank you, I will raise my score accordingly.

---

> > > > ### Author Response · Authors · 2025-08-03
> > > > **Thanks to the Reviewer**
> > > >
> > > > We would like to sincerely thank the Reviewer for their thoughtful and in-depth evaluation of our work. Their positive comments and thoughtful criticism led to improving the quality of our paper.

---

### Official Review · Reviewer_HrsH · 2025-07-07

**Clarity:** 3
**Significance:** 2
**Originality:** 3
**Rating:** 5
**Confidence:** 3

**Summary:**

The authors propose a lightweight method for approximating solutions to SB problems. Their approach has them approximate the boundary distributions by Gaussian Mixtures (GM) and then approximately solving the SB problems for the GMs. Their theoretical results motivate the algorithm for doing the latter, showing that any solution to the GM SB problems has a policy and flow that can be described as a combination of the policy and flow for the SB solution between each of the individual Gaussians. They provide experiments comparing their results to existing SB algorithms, demonstrating improvements.

**Questions:**

- Does the final algorithm optimise J_OT to obtain \lambda_{ij}?
- If so, is it clear how close the objective is to J_GMM or how well the approximation is actually solving the GM SB problem (not just the OT problem). Could you please point to evidence for this?
- Is \rho_{ij} and u_{ij} obtainable in closed-form? If not, how difficult is it to approximate?

**Ethical Concerns:**

["NO or VERY MINOR ethics concerns only"]

**Final Justification:**

The paper proposes a fairly naive methodology, but it does seem to work well. I think its simplicity is a strength and not a weakness and the fact it works well will be interesting to the ML community. Also the quality of presentation, the depth to which the idea was explored and the fact that rigorous theoretical results were provided make this worth accepting.

**Limitations:**

Yes

**Quality:**

3

**Strengths And Weaknesses:**

Strengths
- The algorithm appears to be highly efficient and most SB algoriths are highly inefficient
- The approach of using GMs as the endpoints is simple and yet appears to be quite novel to the literature. The proposal of this could motivate future work
- The paper is largely well-written with the preliminaries section especially serving as a high quality introduction to the field
- The theoretical results regarding SB for GMs is interesting in its own right and could motivate future work
- The extension of the work to momentum SB and continuous GMs demonstrates a deep exploration of the ideas considered in the work.

Weaknesses
- The algorithm they ultimately use is not very clearly presented or discussed much. It would be good to include a clear presentation of this following the theoretical results. For example, it would be good to know how J_OT should be optimised
- The Comparison seems somewhat limited, focussing primarily on comparison with other light methods. A more thorough comparison with standard SB algorithms would help.
- It is not clear to me how close J_GMM is to the proxy J_OT. There are some experiments in table 4 that provide some results in this direction, but a more thorough discussion and set of experiments about how much the proposed method actually solves the GM problem would greatly improve to the validity of the paper.

---

> ### Author Rebuttal · Authors · 2025-07-30
>
> We would like to thank the Reviewer for his/her constructive comments and for the positive evaluation of our work.
> In the following, we address the comments and questions of the Reviewer one by one.
>
> 1. **Optimization of $J_{\\mathrm{OT}}$ to obtain $\lambda_{ij}$**: In order to obtain the weights $\lambda_{ij}$ of the $(i, j)$-GSB subproblem, the algorithm indeed optimizes the cost function $J_{\mathrm{OT}}$ of the linear optimization problem given by Equation (15) in Theorem 2. To solve Problem (15), each term $J_{ij}$ is first computed through Proposition 2 and then, using the known constants $\alpha_0^i, \alpha_1^j$ the linear program (15) is solved using off-the-shelf LP solvers such as MOSEK.
>
> 2. **Bounds on the optimality gap**:  We thank the Reviewer for this question. As the Reviewer points out, some empirical evidence about the optimality gap is presented in Table 4 and also in the extensive benchmark testing in Section C.4.
> To gain a deeper theoretical understanding for the cases when Theorem 2 introduces some sub-optimality, we provide the following alternative proof of Theorem 2, along with a result proving that when the mixture components are well separated, Theorem 2 yields the optimal solution.
>
> Specifically, in the setting of Problem (12), let $\rho_{t|ij}, u_{t|ij}$ be the solution of the $(i,j)$-GSB, with boundary distributions the $i$-th component of the initial and the $j$-th component of the terminal mixture, let $\rho_t = \sum_{ij} \lambda_{ij} \rho_{t|ij}$ and let
> \\[
> u_t(x) = \sum_{i,j} {u_{t|ij}(x) \frac{ \rho_{t|ij}(x)\lambda_{ij}} {\rho_{t}(x)}} = \mathbb{E} [ \omega_t(x)],
> \\]
> where $\omega_{t}(x)$ follows a discrete distribution defined by $\\{ \omega_t(x) = u_{t|ij}(x) \\, \\, \mathrm{w.p.} \\, \\, \frac{ \rho_{t|ij}(x)\lambda_{ij}} {\rho_{t}(x)} \\}$.
>
> Note that for any random variable $x \in \mathbb{R}^n$, the variance decomposition yields
> \\[
>     \\| \mathbb{E}[x] \\|^2 = \mathbb{E}[ \\|x\\|^2] - \mathbb{E}[ \\|x  - \\mathbb{E}[x]\\|^2 ].
> \\]
> Using the last equation and the expression $u_t(x)$ above, written as an expectation, we obtain
>    \\[
>    J_{\mathrm{GMM}} \triangleq  \int_0^1 \\!\\! \int_{\mathbb{R}^n} \\!\\! \rho_{t} (x) \\| u_t(x) \\|^2 \\, \mathrm{d} x \\, \mathrm{d} t
>    \\]
>
>    \\[
>    \Rightarrow J_{\mathrm{GMM}} =\int_0^1 \int_{\mathbb{R}^n}  \sum_{i,j} \lambda_{ij} \rho_{t|i,j} \\|u_{t|i,j}(x)\\|^2  \\, \mathrm{d} x \\, \mathrm{d} t - \int_0^1 \int_{\mathbb{R}^n} \sum_{i,j} \lambda_{ij} \rho_{t|i,j} \\|u_{t|i,j}(x)- u_t(x)\\|^2  \\, \mathrm{d} x \\, \mathrm{d} t
>    \\]
> \\[  \Rightarrow  J_{\mathrm{GMM}} = J_{\mathrm{OT}} - \int_0^1   \int_{\mathbb{R}^n} \sum_{i,j} \lambda_{ij} \rho_{t|ij} \\|u_{t|ij} - u_t \\|^2  \\, \mathrm{d} x \\, \mathrm{d} t  \\]
> The fact that the second term in the last equation is non-negative justifies the upper bound in Theorem 2.
> Next, we show that when the conditional densities are well separated, the second term in the last equation becomes arbitrarily small, and the bound becomes tight.
> Expanding the term inside the norm in the integral of the last equation, and dropping the dependence on $x$ for notational convenience, we obtain
>  \\[ \\|u_{t|i,j} - u_t\\|^2 \\]
>  \\[ = \bigg\\|u_{t|i,j}-  \sum_{i'j'} u_{t|i'j'} \frac{\lambda_{i'j'} \rho_{t|i'j'}}{\rho_t} \bigg\\|^2   \\]
>  \\[ = \bigg\\| \frac{ u_{t|ij} \rho_{t}- \sum_{i'j'} u_{t|i'j'} \lambda_{i'j'} \rho_{t|i'j'} }{ \rho_t} \bigg \\|^2 \\]
> \\[ \leq\left(\sum_{i'j'}\Vert u_{t|i,j}-u_{t|i'j'}\Vert\frac{\lambda_{i'j'} \rho_{t|i'j'}}{\rho_t}\right)^{2} \\]
> \\[ \leq\sum_{i'j'}\Vert u_{t|i,j}-u_{t|i'j'}\Vert^{2}\frac{\lambda_{i'j'} \rho_{t|i'j'}}{\rho_t} \\]
> \\[ \sum_{\\substack{i' \neq i \\ j' \neq j}}\Vert u_{t|i,j}-u_{t|i'j'}\Vert^{2}\frac{\lambda_{i'j'} \rho_{t|i'j'}}{\rho_t} \\]
> where in the 4th equation we applied the discrete version of Jensen's inequality.
> Substituting this upper bound in the expression for $J_{\mathrm{GMM}}$ and rearranging, we get
>
>  \\[ 0 \leq J_{\mathrm{OT}} - J_{\mathrm{GMM}} \\]
>  \\[ \leq \int_0^1 \int_{\mathbb{R}^n} \sum_{i,j} \lambda_{ij} \rho_{t|i,j} \\|u_{t|i,j}(x)- u_t(x) \\|^2  \\, \mathrm{d} x \\, \mathrm{d} t \\]
>  \\[ \leq  \int_0^1  \int_{\mathbb{R}^n} \sum_{i,j}  \sum_{\substack{  i' \neq i \\ j' \neq j} } \big \\|u_{t|ij}  -  u_{t|i'j'} \big \\|^2 \frac{ \lambda_{i'j'} \lambda_{ij} \rho_{t|i'j'} \rho_{t|i,j}  }{  \rho_{t}  }  \\, \mathrm{d} x \\, \mathrm{d} t \\]
>   \\[ \leq  \int_0^1 \int_{\mathbb{R}^n} \sum_{i,j}  \sum_{\substack{  i' \neq i \\ j' \neq j} } \big \\|u_{t|ij} - u_{t|i'j'} \big \\|^2 \min \\{ \lambda_{ij} \rho_{t|ij},  \lambda_{i'j'} \rho_{t|i'j'} \\}  \\, \mathrm{d} x \\, \mathrm{d} t  \\]
> where the last inequailty comes from the inequality $\frac{a_i a_j}{\sum_i a_i} \leq \min\\{a_i, a_j\\}$ for all positive numbers $\\{a_i\\}_{i=1}^N$.
>
> Intuitively, when the components are well separated, the term $\min \\{ \lambda_{ij} \rho_{t|ij},  \lambda_{i'j'} \rho_{t|i'j'} \\}$ becomes arbitrarily small, making the bound tight.
> More rigorously, noting that, for any vectors $a,b$, $\min\\{a,b\\} = \frac{1}{2}(a+b) - \frac{1}{2}|a-b|$,
> we get
>
> \\[ \int_{\mathbb{R}^n} \min \\{ \lambda_{ij} \rho_{t|ij},  \lambda_{i'j'} \rho_{t|i'j'} \\} \\, \mathrm{d} x \\]
> \\[ =\frac{1}{2} (\lambda_{ij} + \lambda_{i'j'}) - \frac{1}{2} \int_{\mathbb{R}^n} | \lambda_{ij} \rho_{t|ij} -  \lambda_{i'j'} \rho_{t|i'j'} | \\, \mathrm{d} x \rightarrow 0 \\]
> or that
>  \\[ \frac{1}{2} (\lambda_{ij} + \lambda_{i'j'}) - \mathrm{TV}(\lambda_{ij} \rho_{t|ij}, \lambda_{i'j'} \rho_{t|i'j'}) \rightarrow 0 \quad \text{as} \quad  \mathrm{TV}(\rho_{t|ij}, \rho_{t|i'j'}) \rightarrow 1 \\]
>
> Since $\int_{\mathbb{R}^n} \min \\{ \lambda_{ij} \rho_{t|ij},  \lambda_{i'j'} \rho_{t|i'j'} \\} \\, \mathrm{d} x\rightarrow 0$ for every $t\in[0,1]$ and
> $\int_{\mathbb{R}^n} \min \\{ \lambda_{ij} \rho_{t|ij},  \lambda_{i'j'} \rho_{t|i'j'} \\} \\, \mathrm{d} x
> \leq\int_{\mathbb{R}^n} \{ \lambda_{ij} \rho_{t|ij}+  \lambda_{i'j'} \rho_{t|i'j'} \} \\, \mathrm{d} x=\lambda_{ij}+\lambda_{i'j'}$
> uniformly in $t$, by bounded convergence theorem, $\int_{0}^{1}\int_{\mathbb{R}^n} \min \\{ \lambda_{ij} \rho_{t|ij},  \lambda_{i'j'} \rho_{t|i'j'} \\} \\, \mathrm{d} x\mathrm{d}t\rightarrow 0$.
>
> Given that the the policies $u_t$ are affine functions of $x$ (see Proposition 1), $\mathbb{E}\_Q [ \\| u_{t|ij}(x) - u_{t|i',j'}(x) \\|^2] < \infty$  where $Q = \\frac{\\min \\{ \lambda_{ij} \rho_{t|ij}, \lambda_{i'j'} \rho_{t|i'j'} \\} }{ \int \int \min \\{ \lambda_{ij} \rho_{t|ij}, \lambda_{i'j'} \rho_{t|i'j'} \\} \\, \mathrm{d} x
> \\, \mathrm{d} t} $, i.e., the normalized distribution of the minimum of the densities $\rho_{t|ij}, \rho_{t|i'j'}$.
> Hence, we conclude that
> \\[ J_{\mathrm{OT}} - J_{\mathrm{GMM}} \rightarrow 0  \quad \text{as} \quad  \mathrm{TV}(\rho_{t|ij}, \rho_{t|i'j'}) \rightarrow 1. \\]
>
> 3. **Closed-form expressions of $\rho_{ij}$ and $u_{ij}$**: Both the policy $u_{t|ij}$, the probability density $\rho_{t|ij}$, and the cost $J_{ij}$ for the $(i, j)$-GSB subproblem can be obtained in closed form through Propositions 1 and 2 for the case of the Gaussian SB. For the more general cases of the Gaussian multi-marginal momentum SB or the more general SB with linear prior dynamics, the policy $u_{t|ij}$, $\rho_{t|ij}$, and $J_{ij}$ are obtained by solving a semidefinite program. We refer to Appendix D for a detailed discussion.
>
>
> 4. **Clear presentation of the algorithm**: We thank the reviewer for this suggestion.
> We agree that an algorithmic block would improve the readability of our work. To this end, we will include a pseudocode block of our method in Section 3 of our manuscript.
> To avoid a duplicate response, we refer the Reviewer to our response to Reviewer VvBB for a preview of our algorithm.
>
>
> 5. **More thorough comparison with standard SB algorithms**: In our original submission, we compared our method against neural approaches in the 2D experiments (Section C.1, Table 4, compared against DSB, DSBM), the EOT benchmark (Section C.4, Table 6, compared against all state-of-the-art methods), and the multi-marginal RNA example (Section 5, Table 3, compared against NLSB, MIOFlow, and DMSB).
> We stress that for the multimarginal momentum SB problem, there are no available light-weight approaches in the literature.
> In the image translation example, we only compared against LSB and LSBM since these match the scope of our algorithm. Due to space and time limitations, we have not compared against neural approaches in this experiment; however, we point the Reviewer to the excellent work [1] for a comparison of LSBM with neural approaches for the image translation task.
>
> [1] Gushchin, N., Kholkin, S., Burnaev, E., \& Korotin, A. (2024, July). Light and optimal schrödinger bridge matching. In Forty-first International Conference on Machine Learning.

---

> > ### Comment · Reviewer_HrsH · 2025-08-05
> >
> > Thank you for the response and for addressing my questions. I will keep my score the same.

---

> > > ### Author Response · Authors · 2025-08-05
> > > **Thanks for the in-depth review.**
> > >
> > > We would like to sincerely thank the Reviewer for their time in providing a high-quality review of our work. We would also like to thank them for their positive comments and for their mindful questions, which led to improving the quality of our paper.

---

### Note · Authors · 2025-08-12

We would like to sincerely thank the reviewers for their careful and positive evaluation of our work. More specifically, we are pleased that the reviewers recognized our paper as well-written (Reviewers HrsH, nHAK, VvBB), pleasant to read (Reviewer PeSU), novel and efficient (all Reviewers), and for finding our preliminaries Section a concise and high-quality introduction to the field (Reviewer HrsH).
In this extensive Rebuttal, we provided detailed answers to all the comments from all Reviewers. **We summarize the main points and improvements we made throughout the rebuttal period here**:

1. To better address questions regarding the optimality of our approach, apart from the empirical study we included in our original submission, which indicated excellent empirical performance of our method, we derived a new result that studies the tightness of the upper bound introduced in Theorem 2. In this direction, we show that when the components of the Gaussian Mixtures are well-separated, the upper bound is tight and solving the optimization problem (15) yields the optimal mixture policy within the feasible set introduced in Theorem 1. The details of this new derivation can be found in point 2. of our response to Reviewer HrsH.

2. We provided new simulations to study the run-times and inference times of our method as the number of components and the problem dimension scale. We also provided an in-depth theoretical complexity analysis study to accompany our empirical results. This further corroborates that our algorithm is very computationally efficient and scales very well in problems with hundreds of components/dimensions. The details of this analysis can be found in our response to Reviewers PeSU and VvBB.

3. To strengthen the presentation of our algorithm, we will include an algorithmic block with a pseudocode outlining our method. A preview of this can be found in point 3. of our response to Reviewer VvBB.

4. Following the suggestion of Reviewer kjHY, in the image translation example, we included a comparative study of the transport costs of all considered methods (GMMflow, LSB, LSBM). A preview of this study can be found in point 4. of our response to Reviewer kjHY.

---

### Decision · Program_Chairs · 2025-09-17

**Decision:**

Accept (spotlight)

**Comment:**

The paper considers the Schrödinger Bridge (SB) problem in two setups: bi-marginal and momentum multi-marginal. For each setting, the authors propose a lightweight algorithm (in the spirit of LightSB work [1]) to obtain an approximate SB solution. The main idea is to approximate marginals with Gaussian Mixture Models (GMMs) and exploit the closed-form SB solutions available between individual Gaussian components.

In the bi-marginal case, the authors first construct a process that is a weighted mixture of pairwise SBs between the components of the two marginals (the weights are optimized). They then construct an "average" process represented by an SDE with an analytically computable drift, made possible by the Gaussian and Gaussian Mixture structure. This SDE is proposed as an approximate SB solution between the mixtures. The multi-marginal case is completely analogous but is more technically complex due to the increased number of marginals.

Overall, the reviewers highly rated the results from the paper: the core idea is interesting, the extension to the multi-marginal case is inspiring (as such extensions are rare), and the resulting algorithm is efficient for low-dimensional problems where GMMs are effective (as demonstrated in experiments). A clear limitation, however, is the method's lack of scalability to high dimensions due to its reliance on GMMs. Furthermore, as noted by several reviewers, there was an initial lack of a clear theoretical understanding of how close the proposed solution is to the true SB. Despite these limitations, the reviewers and meta-reviewer find the paper to be technically solid, original, and well-executed. Its strengths are deemed to outweigh its limitations, and acceptance is recommended.

**P.S.** From the perspective of related work [2], it seems that the proposed algorithm can be viewed as a Markovian projection of a special reciprocal process constructed as a mixture of SBs.

[1] Korotin, A., Gushchin, N., & Burnaev, E. Light Schrödinger Bridge. In The Twelfth International Conference on Learning Representations.

[2] Shi, Y., De Bortoli, V., Campbell, A., & Doucet, A. (2023). Diffusion schrödinger bridge matching. Advances in Neural Information Processing Systems, 36, 62183-62223.